



# Sample labeling and classification method of hyperspectral remote sensing images based on texture features and semi-supervised learning

Ansheng Ye [1,2], Xiangbing Zhou [3,*], Yu Gong [4], Fang Miao [1], Huimin Zhao [4,*]

[1] *Key Lab of Earth Exploration & Information Techniques of Ministry Education, Chengdu University of Technology,*
*Chengdu 610059, China*

[2]*School of Computer Science, Chengdu University, Chengdu 610106, China*

[3]*School of Information and Engineering, Sichuan Tourism University, Chengdu 610100, China*

[4] *College of Electronic Information and Automation, Civil Aviation University of China, Tianjin 300300, China*

*Corresponding author: zhouxb@uestc.edu.cn (Xiangbing Zhou);   hm_zhao1977@126.com (Huimin Zhao)

## Abstract

Hyperspectral images contain abundant spectral and spatial information about the earth's surface, labeling data processing and analysis more difficult, as well as the problem of sample labeling. In this paper, local binary pattern (LBP), sparse representation and mixed logistic regression model are introduced, and a sample labeling method based on neighborhood information and priority classifier discrimination is presented. Then, a hyperspectral remote sensing image classification method based on texture features and semi-supervised learning is implemented. The LBP is employed to extract features of spatial texture information from remote sensing images and enrich the feature information of samples. Then the multivariate logistic regression model is used to select the unlabeled samples with the largest amount of information, and the unlabeled samples with neighborhood information and priority classifier tags are selected to obtain the pseudo-labeled samples after learning. By making full use of the advantages of sparse representation and mixed logistic regression model, a new hyperspectral remote sensing image classification model based on semi-supervised learning is constructed to effectively achieve accurate classification of hyperspectral images. The data of Indian Pines, Salinas scene and Pavia University are selected to verify the validity of the proposed method. The experiment results show that the proposed classification method can obtain higher classification accuracy and show stronger timeliness and generalization ability.

*Keywords:* Hyperspectral remote sensing image; Local binary pattern; Sparse representation; Mixed logistic regression; Neighbourhood information

## 1. Introduction

Hyperspectral Image (HSI) is the simultaneous imaging of target areas in dozens to hundreds of continuous spectral bands. It effectively integrates the spatial and spectral information in the imaging scene, with strong target detection ability and better material identification ability (Chang et al., 2021; Chen et al., 2021; Dou et ai., 2020). It is widely used in agriculture and forestry, geological exploration, marine exploration, Environmental monitoring and other fields. However, HSI is characterized by high data dimension, large information redundancy and high correlation between bands, which brings great difficulties to its processing and classification (Dumke et al.,2019; Huang et al., 2020; Jiang et al., 2020; Seifi et al., 2017). Therefore, how to reduce the redundant information of the data, extract and use the features of the hyperspectral image effectively, and realize the accurate classification of the hyperspectral image are the hot and difficult issues in the current hyperspectral image processing and classification research.

Sample labeling of hyperspectral image data often requires expert knowledge and experience, so the cost of sample labeling is high(Shang et al., 2020). When the labeled samples are limited, semi-supervised learning can explore the useful information of the unlabeled samples to participate in the model training and reduce the labeling cost(Shi et al., 2019; Ye et al., 2021). In the field of machine learning, semi-supervised learning acquires knowledge and experience from a small number of labeled samples. Mining usable information from a large number of unlabeled samples helps the classification model to train and improve the classification accuracy (Yin et al., 2021; Yu et al., 2021; Chen et al., 2020 ). Therefore, a large number of scholars have carried out the research of semi-supervised learning in remote sensing images. Camps-Valls et al. (Camps-Valls et al., 2007) proposed a graph-based hyperspectral image classification method, and constructed the graph structure through the graph



method. The data context information is integrated based on the composite kernel and the Nystrom method is introduced to
speed up classification. Yang et al. (Yang et al., 2012) proposed a semi-supervised band selection technique for hyperspectral
image classification. A metric learning method is used to measure the features of hyperspectral images, and a semi-supervised
learning method is used to select a subset of valid bands from the original bands. The validity of the method and the
improvement of classification accuracy are verified by experiments. Tan et al. (Tan et al., 2014) proposed a hyperspectral image
classification method based on segmentation integration and semi-supervised support vector machine. The spatial information
of the tag samples is extracted using a segmentation algorithm to filter the samples, and then classified based on semi-supervised
learning. Samiappan et al. (Samiappan et al., 2015) combined active learning and co-training to perform semi-supervised
classification of hyperspectral images. The initial classification model is trained according to the labeled samples, and the
heuristic active learning is performed on the unlabeled samples. Combined with the original data, the labeled sample set was
divided into views, and the unlabeled samples with high heuristic values were selected to join the training sample set for co-
training. Zhang (Zhang et al., 2016) used a semi-supervised classification method based on hierarchical segmentation and active
learning to extract spatial information from hyperspectral images, then the training set is updated iteratively by using the
information of a large number of unlabeled samples to complete the hyperspectral image classification.

15       In hyperspectral images, each pixel corresponds to a spectral curve that reflects its inherent physical, chemical and optical

properties. The main basis of hyperspectral image classification is to use the feature information of different pixels to label the
pixels belonging to different landmarks and obtain the corresponding classification maps (Zhang et al., 2022; Zhao et al., 2022).
Therefore, a large number of scholars have carried out the research on hyperspectral image classification. Melgani et al.
(Melgani et al., 2004) proposed a hyperspectral image classification method based on Support Vector Machines (SVM). The
kernel function is introduced to solve the nonlinear separable problem and avoid the curse of dimensionality. Ratle et al. (Ratle
et al., 2006) introduced neural networks into hyperspectral image classification. In the training phase, the loss function is
optimized to avoid problems such as local optimization. Chen et al. (Chen et al., 2011) constructed a hyperspectral image
classification model based on sparse representation, and compared the classification results of common machine learning
methods. In order to improve the shortcomings of sparse representation in dealing with nonlinear problems, Chen et al. (Chen
et al., 2013) introduced kernel method to propose a kernel sparse representation technique. In addition, Cui et al. (Cui et al.,
2013) proposed a multiscale sparse representation algorithm for robust hyperspectral image classification. Automatic and
adaptive weight allocation schemes based on spectral angle ratio are incorporated into the multi-classifier framework to fuse
sparse representation information at all scales. Tang et al. (Tang et al., 2016) proposed two sparse representation algorithms
based on manifolds to solve the instability problem of l1-based sparse algorithms. Using regularization and local invariance
techniques, two manifold-based regularization items are merged into the$l_1$-based objective function. Wang et al. (Wang et al.,
2016) applied the neighborhood-cutting technique to sparse representation, and combined the joint spatial and spectral sparse
representation classification algorithm. Wang and Celik (Wang et al., 2018) improved the classification accuracy of
hyperspectral images by combining context information in the sparse coefficient domain. Hu et al. (Hu et al., 2019) proposed
two weighted kernel joint sparse representation methods, which determine the calculation weight by calculating the kernel
similarity between adjacent pixels. The nearest neighbor regularization strategy is used to optimize both the weight of the
projected adjacent pixels and the joint sparse representation factor. Xue et al. (Xue et al., 2017) presented two novel sparse
graph regularization methods, SGR and SGR with total variation. Yang et al. (Yang et al., 2018) studied the effect of the p-
norm distance metric on the minimum distance technique and proposes a supervised-learning p-norm distance metric to
optimize the value of p. Zhang et al. (Zang et al., 2019) proposed a multi-scale dense network for HSI classification that made
full use of different scale information in the network structure and combined scale information throughout the network. Liu et
al. (Liu et al., 2021) proposed a class-wise adversarial adaptation in conjunction with the class-wise probability MMD as the
class-wise distribution adaptation (CDA) network. Wang et al. (Wang et al., 2022) proposed graph-based semi-supervised
learning with weighted features for HSI classification.

44       To sum up, hyperspectral images contain rich spectral and spatial information of earth surface features, which increases the

difficulty of data processing and analysis. In addition, the training samples of actual hyperspectral images are small and there
is a problem of sample labeling. The local binary pattern, sparse representation and mixed logistic regression model are used

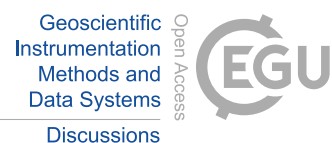

in this paper. A new hyperspectral image feature extraction method based on local binary pattern is proposed to obtain texture
features of hyperspectral image samples and enrich hyperspectral image sample information. A sample selection strategy based
on active learning is designed to determine the unlabeled samples. Based on this, a new sample labeling method based on
neighbourhood information and priority classifier discrimination is deeply studied to expand the training samples. The
hyperspectral remote sensing image classification method based on texture features and semi-supervised learning is studied to
improve the classification accuracy of remote sensing images.
The main contributions of this paper are described as follows.
1) A novel a hyperspectral remote sensing image classification method based on texture features and semi-supervised
learning is proposed, which introduces local binary pattern, sparse representation, hybrid logistic regression model and so on.
2) The local binary pattern is used to effectively extract the features of spatial texture information of remote sensing images
and enrich the feature information of samples.
3) A multiple logistic regression model was used to optimally select unlabeled samples, which are labeled by using
neighbourhood information and priority classifier discrimination to achieve pseudo-labeling of unlabeled samples.
4) A hyperspectral remote sensing image classification model based on semi-supervised learning is constructed to effectively
achieve accurate classification of hyperspectral images by making full use of the advantages of sparse representation and mixed
logistic regression model.
**2. Basic methods**
*2.1. Local binary pattern (LBP)*
LBP is a feature extraction method that extracts spatial texture information of images. Texture, which is widely used
in image processing and image analysis, represents the slow change or periodic change of the surface structure of the
object(Ojala et al. 1996). LBP is also widely used in feature extraction of hyperspectral images due to the simple structure
and easy calculation. Give the center pixel $g_c(x_c, y_c)$ and the neighborhood pixel $g_p$,
$$g_p = (x_c + Rcos\left(\frac{2\Pi p}{P}\right), y_c - Rsin(\frac{2\Pi p}{P})) \tag{1}$$

where, $g_p(p=0,1,...,P-1)$ represents the coordinate values of P pixels uniformly distributed on the circular
domain with $g_c$ as the centre and R as the radius. The local texture information at the center pixel is the circular area
in the Figure 1, which can be represented.

| 7 |  | ... |  | 28 |
|---|---|---|---|---|
|  | 79 | 26 | 78 |  |
|  | 132 | 68 | 10 |  |
|  | 30 | 202 | 252 |  |
| 24 |  | ... |  | 59 |

**Figure1.** The quantized texture feature form of one region
$$LBP_{g_c} = 2^p \times \sum_{p=0}^{P-1} s(g_p - g_c) \tag{2}$$

$$s(x) = \begin{cases} 1, x > 0 \\ 0, x \le 0 \end{cases} \tag{3}$$

*2.2. Sparse expression*
Sparse representation means that the signal can be approximately represented by a linear combination of the atoms in the
dictionary. Now, $X = [X_1, X_2,...,X_c] \in R^D$ is given as the HSI pixel and $D$ is the number of image bands. In here,
$X_i = [x_{i1}, x_{i2},...,x_{iN_c}] \in R^D$, $N_c$ represents the number of samples in class $i$.



For samples in class $i$ , it can be approximated as follow.

$$y \approx x_{i1}\alpha_1 + x_{i2}\alpha_2 + ... + x_{iN_c}\alpha_{N_c}$$

2                    $= \left[ x_{i1}, x_{i2}, ..., x_{iN_c} \right]\left[ \alpha_1, \alpha_2, ..., \alpha_{N_c} \right]^T$         (4)

                    $= X_i\alpha_i$

where, $X_i$ represents a sparse sub-dictionary of the samples in class $i$ . $\alpha_i$ represents the sparse vector of test samples $y$ ,
which contains only a few non-zero values.
In order to obtain the sparsest vector $\alpha_i$ , the following formula is solved.
$$\tilde{\alpha} = \arg\min \|\alpha_i\|_0 , s.t. y = A\alpha_i \qquad (5)$$
where, $\| . \|_0$ is a $l_0$ norm, which represents the number of non-zero atoms in the vector, also known as sparsity. $A$ is a
sparse dictionary. It is a NP-hard problem to solve the formula directly. Under some conditions, the minimization solving
problem($l_0$) is approximated by the minimization solving problem($l_1$ ), which can be relaxed.
$$\tilde{\alpha} = \arg\min \|\alpha_i\|_1 , s.t. y = A\alpha_i \qquad (6)$$
Furthermore, the solution can be converted to the following formula.
$$\tilde{\alpha} = \arg\min \|\alpha_i\|_0 , s.t. \|A\alpha_i - y\|_2^2 < \varepsilon \qquad (7)$$
where, $\varepsilon$ represents the refactoring error. Orthogonal matching pursuit (OMP) algorithm can be used to solve the above
equation. After the sparsity coefficient is calculated, the reconstruction residual for each class of the test sample $y$ can be
calculated.
$$r_i(y) = \|y - A\tilde{\alpha}_i\|_2 \qquad (8)$$
where, $i \in \{1, 2, ..., C\}$ .
Finally, the reconstruction residuals of all category dictionaries are compared. The minimum residual is the category y.
$$class(y) = \arg\min(r_i(y)), i = 1, 2, ... C \qquad (9)$$
**3. An image sample labeling method based on neighbourhood information and priority classifier discrimination**
*3.1. Sample selection method based on multiple logistic regression model*
Before the samples are labeled, sample selection is required. This is because if all unlabeled samples are labeled directly, all
unlabeled samples need to be labeled, which will cost a lot of computational cost. Moreover, due to the small number of initial
labeled samples and the limited information available, it is difficult to label some samples with a certain accuracy. Mislabeled
samples obviously affect the classification accuracy of the model. The main objective of the sample selection strategy is to
select the unlabeled samples with the largest amount of information. These unlabeled samples can construct a valuable training
set after labeling, and effectively promote the improvement of classification results. Therefore, a kind of information selection
method based on multiple logistic regression model is proposed to realize the selection of samples.
That is to say, the classified probability matrix $p(y_i^k | x_i)$ of each sample by using multiple logistic regression model has a
large amount of information that can be mined as the initial data. The multiple logistic regression classifier is modeled by
discriminant Bayesian decision model. According to the theory of generalized linear model, it can be obtained as follow.
$$P(y; \delta) = b(y)\exp\left(\delta^T T(y) - a(\delta)\right) \qquad (10)$$
The specific form of multiple logistic regression is described as follow.
$$p(y_i = k | x_i, \eta) = \exp\left(\eta^k g(x_i)\right) \Big/ \sum_{k=1}^{N} \exp\left(\eta^k g(x_i)\right) \qquad (11)$$
where, $g(x) = [g_1(x), g_2(x), ..., g_f(x)]^T$ is the feature vectors of the input, and $\eta = [\eta_1^T, \eta_2^T, ... \eta_k^T]$ represents the
regression parameter vector of the classifier. It is worth noting that the feature vector is often represented by introducing the





idea of kernel, which is not only used to improve the indivisibility, but also helps the classifier to fit better by training samples.
Generally, the kernel function is radial basis function (RBF) as follow.

$$K(x_m, x_n) = e^{\frac{-||x_m - x_n||^2}{2p^2}}$$
(12)

After the feature vector is determined, the regression parameter $\eta$ of model is only determined, and then the probability
matrix $p(y_i^k|x_i)$ of each unlabeled sample belonging to each class is determined. The amount of information of the samples
is determined by the Breaking Ties (BT) and the Least Confidence (LC). In this paper, the BT method is selected to determine
the amount of information.
The BT method shows the similarity between the two categories by comparing the difference between the maximum category
probability and the sub-maximum category probability. The difference is smaller, the similarity between the two types of
samples is greater. The uncertainty is greater, the amount of information is greater. $S_i$ is used to indicate the similarity between
categories. The formula is described as follow.

$$S_i = maxp(y_i^k|x_i) - secondmax(p(y_i^k|x_i))$$
(13)

The $S_i$ is finally sorted in ascending order.
*3.2. A sample labeling method based on neighborhood information and priority classifier*
The features of hyperspectral images have some correlation. The ground objects are closer, the correlation is stronger. In the
research of sample labeling, spatial neighborhood information based on training samples is widely used. However, due to the
unknown central pixel and the lack of sufficient determination information, the neighborhood information of unlabeled samples
is relatively less in the research of sample labeling. Generally, the label of any pixel on a hyperspectral image must be consistent
with the label of one pixel in its neighborhood. This property can be applied to label the unlabeled samples. The label
information of training samples around the unlabeled samples can be used to discriminate the unlabeled samples. The labeling
discrimination method based on neighborhood information centers on the sample to be labeled. The labeled samples appearing
around it are labeled with a block diagram. All the occurrences of sample labels are recorded and denoted as the neighborhood
information set. Then, the labeled samples are used as training samples to train the classifier and classify the unlabeled samples.
Determine whether the predicted sample label by the classifier appears in the neighborhood information set of the unlabeled
samples. If it appears, the predicted label by the classifier is the sample label. Otherwise, the samples are put to be labeled back
into the unlabeled sample set. One of the most important problems is whether the unlabeled samples which satisfy the
neighborhood information can be reliably labeled by the classifier. At present, some studies use multiple classifiers to
discriminate together and achieve good classification effect. However, a problem is how to determine the determination of
labels, when the predicted labels by multiple classifiers are inconsistent, but all appear in the neighborhood information set of
unlabeled samples.
Therefore, a sample labeling method based on priority classifier discrimination is proposed in this paper. For unlabeled
samples with the neighborhood information, the classifier with the highest priority is used for prediction. If the obtained
prediction marker appears in the neighborhood information set, its marker is determined. Otherwise, the classifier with the
lowest priority is used for prediction. Then judge whether the label can be determined until the end of the sample labeling. The
sample labeling method based on neighborhood information and priority classifier discrimination is shown in Figure 2.

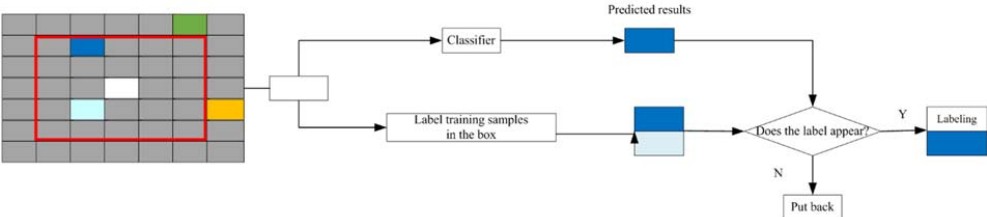

**Figure 2.** Sample labeling process based on neighborhood information and priority classifier discrimination
This labeling method is a cyclical iterative process. Although it is not possible to ensure enough training samples around all
unlabeled samples at the initial stage of sample labeling, it can ensure that some unlabeled samples are sufficient. The unlabeled


samples are then labeled and extended to the training set. With each iteration, the training set grows. Those unlabeled samples
whose neighborhood training samples are not sufficient may reach the label condition at a certain labeling time. This sample
labeling method with replacement ensures the accuracy of sample labeling to a certain extent, and improves the performance
of classifier step by step.
**4. Hyperspectral image classification method based on texture features and semi-supervised learning**
*4.1. The idea of hyperspectral image classification*
Hyperspectral images consist of pairs of continuous spectral bands, which contain rich spectral and spatial information of
earth surface features. So that some objects that cannot be identified by conventional remote sensing means can be identified
in hyperspectral images. However, the abundant data information increases the difficulty of data processing and analysis, and
there are problems such as the difficulty of sample labeling. In order to improve the accuracy of hyperspectral image
classification, a new hyperspectral image classification method based on texture features and semi-supervised learning is
proposed in this paper. Firstly, aiming at the problems of high correlation between bands, information redundancy, high data
dimension and complex processing, LBP is employed to deal with the hyperspectral images. The texture features of
hyperspectral images are effectively extracted to enrich the feature information of samples. Then, to solve the problem of
limited label samples, a new sample labeling method based on neighborhood information and priority classifier is proposed.
And a sample selection strategy is designed to find some samples from a large number of unlabeled samples. Secondly, the
selection samples are labeled by using the neighborhood information and the priority classifier. Finally, the classifier is applied
to achieve accurate classification of hyperspectral images.
*4.2. The model of hyperspectral image classification*
The hyperspectral image classification model based on texture features and semi-supervised learning is shown in Figure 3.

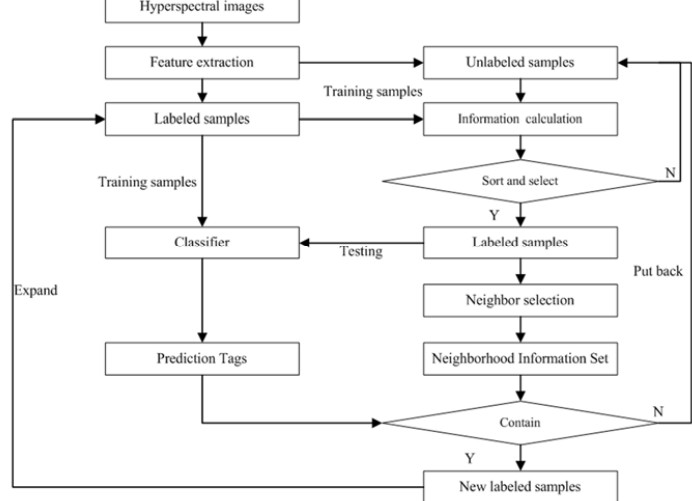

**Figure 3.** Hyperspectral image classification model based on texture features and semi-supervised learning
**5. Case analysis**
*5.1. Experimental data*
(1) Indian Pines data
The images of Indian pines in northwest Indiana were collected by AVIRIS sensor. The images consist of $145 \times 145$ pixels
and 224 spectral reflection bands with a wavelength range of 0.4~2.5 nm, including 16 types of feature elements. The false
color map and real ground object distribution are shown in Figure 4.





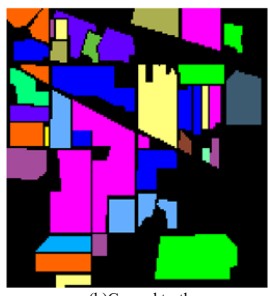

(a)False color image      (b)Ground truth

**Figure 4** Hyperspectral remote sensing images of Indian Pines

(2) Salinas Scene data

The AVIRIS spectrometer collects images of the Salinas Valley in California, USA, with a size of $512 \times 217$ pixels and a total of 224 bands. After removing the bands covering the water absorption area, 204 bands were used, including 16 types of ground feature elements. The false color map and the real ground object distribution are shown in Figure 5.

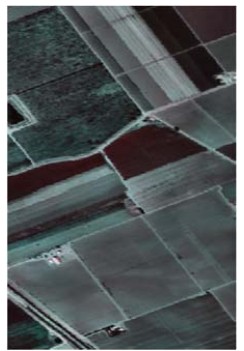
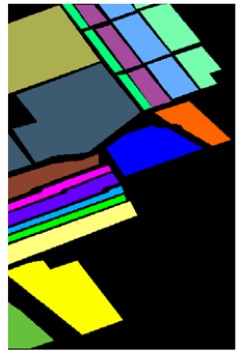

(a)False color images      (b)Ground truth

**Figure 5** Hyperspectral remote sensing image of Salinas Scene

(3) Pavia University data

Images of the Italian University of Pavia campus taken by the Rosis Spectrometer. It is $610 \times 340$ pixels in size and has a total of 115 wavebands. The 103 wavebands after removing the wavebands covering the water-absorbing region contain a total of 9 types of features. The false color map and the real ground object distribution are shown in Figure 6.

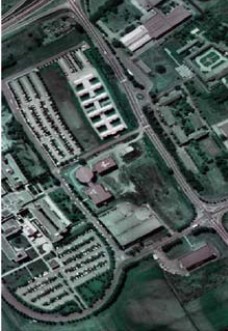
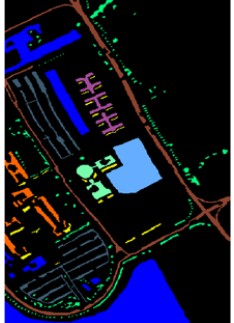

(a)False color image      (b)Ground truth

**Figure 6** Hyperspectral remote sensing image of Pavia University

In the experiment, 10% of each type of ground object of the three kinds of data is randomly selected as the training samples, and the rest is the test samples.

*5.2. Evaluation criteria*
Confusion Matrix (CM) is usually used in the classification and evaluation of hyperspectral images. A confusion matrix is
generally defined as follow.

$$P = \begin{bmatrix} p_{11} & p_{12} & \cdots & p_{1n} \\ p_{21} & p_{22} & \cdots & p_{2n} \\ \cdots & \cdots & \cdots & \cdots \\ p_{n1} & p_{n2} & \cdots & p_{nn} \end{bmatrix} \tag{14}$$

where, n denotes the number of objects in the category, $p_{ij}$ represents the number of samples belonging to class i that were
assigned to class j. The total amount of data in each row denotes the true number of objects in that category. The total amount
of data for each column represents the total number of samples.
Based on the confusion matrix, three classification indexes can be obtained, which are Overall Accuracy (OA), Average
Accuracy (AA) and Kappa coefficient.

$$OA = \frac{\sum_{i=1}^{n} p_{ii}}{N} \tag{15}$$

where, N represents the total number of samples participating in the classification. $p_{ii}$ represents the number of correctly
classified samples of class i. It represents the probability that the classified result corresponds to its true label for each random
sample.

$$CA_i = \frac{p_{ii}}{N_i} \tag{16}$$

where, $N_i$ represents the total number of samples for the first category in class i. $CA_i$ represents the probability that
category i is correctly classified.

$$Kappa = \frac{\left(n\left(\sum_{i=1}^{N} p_{ii}\right) - \sum_{i=1}^{N}\left(\sum_{j=1}^{N} p_{ij} \sum_{j=1}^{N} p_{ji}\right)\right)}{n^2 - \sum_{i=1}^{N}\left(\sum_{j=1}^{N} p_{ij} \sum_{j=1}^{N} p_{ji}\right)} \tag{17}$$

The Kappa coefficient comprehensively considers the number of objects correctly classified and the error of being
misclassified on the diagonal of the confusion matrix.
*5.3. Parameter determination and analysis*
*5.3.1 Sample selection methods*
The quality of sample selection directly affects the efficiency of the whole experiment, and also affects the performance of
classifier. In order to select the best sample selection method, the experimental results of IE, ME, BT and LC on three kinds of
hyperspectral images were compared. The experiment is set to take 10 initial samples for each class, and all remaining samples
are test samples. Two hundred unlabeled samples were selected by four sample selection methods in each iteration, and the
real labels were given to the unlabeled samples. The samples were used to expand the training set, train the classifier and
classify the test samples. The quality of the sample selection method is determined according to the classification results after
each iteration is executed. The classification accuracy of different sample selection methods on different data sets is shown in
Table 1.

**Table 1** The classification accuracy of different methods on different data sets(%)

| Data | Selection method | 1 | 2 | 3 | 4 | 5 | 6 | 7 | 8 | 9 | 10 |
|---|---|---|---|---|---|---|---|---|---|---|---|
| Indian Pines | IE | 78.39 | 79.10 | 79.99 | 80.71 | 82.11 | 83.47 | 84.10 | 85.51 | 86.48 | 87.23 |
| | ME | 80.33 | 86.83 | 91.08 | 92.92 | 95.28 | 97.02 | 98.00 | 98.45 | 98.90 | 99.13 |
| | BT | **91.47** | **95.47** | **98.22** | **98.36** | **98.64** | **98.59** | **98.66** | **98.71** | **99.34** | **99.29** |
| | LC | 84.74 | 88.95 | 91.63 | 94.33 | 95.27 | 96.46 | 98.15 | 98.55 | 98.62 | 98.66 |
| Pavia University | IE | 69.87 | 71.31 | 71.75 | 71.74 | 72.04 | 72.40 | 73.25 | 73.36 | 73.76 | 74.45 |
| | ME | 73.01 | 75.23 | 78.74 | 82.78 | 89.14 | 92.82 | 95.14 | 96.10 | **97.13** | **97.82** |
| | BT | **87.54** | **92.63** | **94.51** | **95.24** | **95.84** | **96.10** | **96.39** | **96.50** | 96.69 | 96.71 |
| | LC | 74.91 | 76.33 | 80.76 | 84.45 | 87.23 | 89.89 | 90.27 | 90.67 | 90.56 | 91.47 |
| Salinas Scene | IE | 84.16 | 84.60 | 84.65 | 85.02 | 85.09 | 85.25 | 85.50 | 85.65 | 85.91 | 85.98 |
| | ME | 85.14 | 89.29 | 92.81 | 94.31 | 96.48 | 97.41 | 98.04 | 98.20 | 98.66 | 98.88 |





| | 1 | 2 | 3 | 4 | 5 | 6 | 7 | 8 | 9 | 10 |
|---|---|---|---|---|---|---|---|---|---|---|
| BT | **95.30** | **96.98** | **98.26** | **98.71** | **98.95** | **98.90** | **99.03** | **99.21** | **99.25** | **99.24** |
| LC | 88.60 | 91.19 | 92.84 | 93.58 | 93.80 | 95.74 | 97.56 | 97.72 | 98.56 | 98.86 |

From Table 1, it can be seen that the classification accuracies of ME, BT and LC are greatly improved with the increase of
the number of iterations on the three data sets. The results of the BT method are significantly better than those of the other
methods. The accuracy can be improved to a high level in the first few iterations, indicating that BT method can select samples
with greater classification improvement. Therefore, the BT method is chosen as the sample selection method in this paper.
*5.3.2 Determination of sample size*
In the labeling process, the samples are not completely labeled correctly. The more samples are screened, the more samples
may be misclassified. This will make the training set more noisy and affects the generalization ability of the classifier. If the
number of samples is too small, the number of labeled samples will not be enough to improve the classification accuracy of the
classifier or will reduce the classification efficiency. The results of classification accuracy under different sample sizes are
shown in Table 2.

**Table 2** The classification accuracy results under different sample sizes(%)

| Data | Quantity | 1 | 2 | 3 | 4 | 5 | 6 | 7 | 8 | 9 | 10 |
|---|---|---|---|---|---|---|---|---|---|---|---|
| Indian Pines | 200 | 77.30 | 77.57 | 78.32 | 77.97 | 78.59 | 78.97 | 78.96 | 79.28 | **79.51** | 79.23 |
| | 400 | 77.54 | 78.60 | 79.80 | 80.80 | 82.13 | 82.20 | 83.03 | 83.77 | 83.83 | **83.85** |
| | 600 | 77.52 | 79.54 | 79.37 | 79.27 | 80.08 | 81.99 | 83.89 | 83.60 | 84.30 | **84.48** |
| | 800 | 77.75 | 79.84 | 80.87 | 80.22 | 82.11 | 83.91 | 84.41 | 84.57 | 85.12 | **85.94** |
| | 1000 | 77.85 | 80.49 | 80.28 | 82.74 | 81.35 | 82.60 | 84.18 | 85.88 | 86.77 | **87.84** |
| | 1200 | 77.85 | 79.95 | 79.79 | 80.38 | 81.59 | 84.41 | 84.72 | 85.74 | 87.48 | **88.85** |
| | 1400 | 78.18 | 80.20 | 80.09 | 83.96 | 85.00 | 85.78 | 87.93 | 89.90 | 91.07 | **91.20** |
| | 1600 | 78.55 | 80.56 | 80.59 | 84.12 | 86.90 | 87.82 | 89.02 | 90.87 | 91.49 | **91.83** |
| | 1800 | 78.34 | 79.76 | 79.23 | 82.64 | 85.39 | 87.32 | 88.81 | 89.97 | 90.69 | **91.46** |
| | 2000 | 78.01 | 79.16 | 80.24 | 82.55 | 86.02 | 87.48 | 88.66 | 89.63 | 90.02 | **90.46** |
| Pavia University | 200 | 68.75 | 73.93 | 76.73 | 78.18 | 79.23 | 80.92 | 81.85 | 82.40 | 82.74 | **83.60** |
| | 400 | 66.41 | 73.11 | 75.45 | 78.20 | 81.18 | 82.13 | 82.57 | 83.39 | **84.07** | 83.98 |
| | 600 | 68.88 | 76.35 | 78.22 | 80.73 | 82.59 | 83.29 | 83.68 | 84.57 | 84.95 | **84.98** |
| | 800 | 69.89 | 77.50 | 80.31 | 81.91 | 83.22 | 84.85 | 84.99 | 84.79 | 85.16 | **85.31** |
| | 1000 | 70.28 | 76.35 | 79.92 | 82.68 | 83.83 | 84.48 | 84.84 | 85.21 | **85.37** | 85.04 |
| | 1200 | 70.24 | 75.18 | 80.32 | 83.13 | 84.14 | 85.04 | 85.30 | **85.55** | 85.04 | 84.90 |
| | 1400 | 70.34 | 76.23 | 80.57 | 82.46 | 83.92 | 84.71 | 85.59 | 85.77 | 85.87 | **86.47** |
| | 1600 | 70.40 | 75.87 | 80.64 | 83.06 | 83.93 | 84.69 | 85.28 | 86.02 | **86.19** | 85.83 |
| | 1800 | 69.77 | 76.12 | 80.18 | 82.99 | 85.19 | 85.04 | 84.89 | 85.26 | **85.68** | 85.68 |
| | 2000 | 69.71 | 75.90 | 82.29 | 83.40 | 84.41 | 85.23 | 85.59 | 85.77 | 85.86 | **85.87** |
| Salinas Scene | 200 | 85.09 | 87.26 | 89.04 | **89.35** | 89.24 | 89.22 | 88.85 | 88.47 | 88.14 | 88.03 |
| | 400 | 84.94 | 88.34 | **89.88** | 89.26 | 89.12 | 88.88 | 88.26 | 87.68 | 87.40 | 87.25 |
| | 600 | 85.69 | **90.85** | 90.80 | 90.42 | 89.71 | 89.04 | 88.63 | 87.84 | 87.12 | 86.60 |
| | 800 | 85.36 | **89.17** | 88.87 | 87.96 | 87.46 | 86.85 | 86.42 | 85.69 | 85.13 | 84.58 |
| | 1000 | 85.35 | 88.93 | 89.88 | **89.04** | 88.34 | 87.58 | 87.03 | 85.49 | 85.08 | 85.08 |
| | 1200 | 85.04 | 89.66 | **90.08** | 88.98 | 87.67 | 87.14 | 86.13 | 85.30 | 85.09 | 84.85 |
| | 1400 | 85.07 | 88.46 | **89.39** | 88.63 | 88.10 | 88.06 | 87.34 | 86.81 | 86.69 | 86.58 |
| | 1600 | 85.47 | 89.54 | **89.88** | 88.87 | 87.68 | 86.21 | 85.00 | 84.34 | 83.89 | 83.70 |
| | 1800 | 85.50 | **90.16** | 90.15 | 89.45 | 88.58 | 87.71 | 86.79 | 86.52 | 86.31 | 85.93 |
| | 2000 | 85.48 | **89.40** | 89.38 | 88.60 | 87.73 | 85.98 | 85.67 | 85.12 | 85.39 | 85.34 |





It can be seen from Table 2 that the selection of sample screening quantity in different data sets presents different rules.
Indian Pines datasets have the highest classification accuracy after 10 iterations are finished. Pavia University datasets
has the highest classification accuracy after 8,9, and 10 iterations are finished. Salinas Scene datasets have the highest
classification accuracy after 2,3, and 4 iterations are finished. The sample screening quantity with the highest accuracy
is regarded as the experimental parameter, which were 1600 for Indian Pines, 1400 for Pavia University and 600 for
Salinas Scene.
*5.3.3 Determination of block window size*
The size of the block window determines the neighborhood information set of the samples, which directly affects the
accuracy of the pseudo-tagging method. Due to the different scale of data sets, the optimal block window size is also
determined by a large number of experiments. The classification accuracy under different block window sizes is shown
in table 3.

**Table 3** The classification accuracy (%) under different block window sizes

| Data | Block window size | 1 | 2 | 3 | 4 | 5 | 6 | 7 | 8 | 9 | 10 |
|---|---|---|---|---|---|---|---|---|---|---|---|
| Indian Pines | 3 | 77.62 | 77.86 | 78.35 | 78.65 | 78.91 | 79.14 | 79.09 | 79.12 | **79.56** | **79.56** |
| | 4 | 77.73 | 78.24 | 78.65 | 78.75 | 78.70 | 78.66 | 79.14 | 79.18 | 79.70 | **79.74** |
| | 5 | 78.41 | 79.61 | 79.41 | 81.28 | 81.11 | 81.73 | 82.67 | 82.84 | 84.18 | **84.68** |
| | 6 | 77.85 | 80.14 | 80.41 | 80.91 | 80.90 | 83.54 | 84.91 | 85.09 | 86.42 | **88.30** |
| | **7** | 78.86 | 80.58 | 82.58 | 84.90 | 86.55 | 86.99 | 88.47 | 88.87 | 89.58 | **90.98** |
| | 8 | 78.33 | 78.95 | 81.00 | 84.24 | 85.44 | 86.37 | 87.60 | 88.13 | 88.57 | **89.19** |
| | 9 | 79.61 | 81.00 | 83.39 | 85.51 | 85.71 | 86.34 | 86.78 | 86.86 | 87.07 | **87.86** |
| | 10 | 78.61 | 79.32 | 83.45 | 85.45 | 85.59 | 85.69 | 86.12 | 87.00 | 87.17 | **87.32** |
| Pavia University | 5 | 68.39 | 68.38 | 68.38 | 68.38 | 68.38 | 68.38 | 68.38 | 68.38 | 68.38 | 68.38 |
| | 10 | 68.20 | 67.67 | 69.05 | 69.06 | 68.30 | 68.13 | 68.06 | 67.98 | 67.89 | 67.62 |
| | 15 | 70.91 | 70.59 | 70.96 | 70.50 | 71.89 | 73.54 | **73.80** | 74.58 | 75.25 | 75.32 |
| | 20 | 71.47 | 71.63 | 73.60 | 76.12 | 75.99 | 75.61 | 77.18 | 77.51 | 77.81 | **78.14** |
| | **25** | 71.25 | 74.57 | 78.52 | 79.15 | 81.95 | 82.32 | 83.66 | 84.88 | 85.45 | **85.51** |
| | 30 | 70.28 | 76.35 | 79.92 | 82.68 | 83.83 | 84.48 | 84.84 | 85.21 | **85.37** | 85.04 |
| | 35 | 71.48 | 77.43 | 80.15 | 81.75 | 83.18 | 83.62 | **84.11** | 83.65 | 83.23 | 83.17 |
| | 40 | 73.40 | 77.50 | 80.36 | 81.92 | **83.15** | 82.68 | 82.23 | 82.38 | 82.18 | 82.04 |
| Salinas Scene | 5 | 83.94 | 83.94 | 83.94 | 83.94 | 83.94 | 83.94 | 83.94 | 83.94 | 83.94 | 83.94 |
| | 10 | 83.61 | 84.16 | 84.95 | 85.03 | 84.77 | 84.68 | 84.59 | 84.82 | 84.61 | **85.48** |
| | 15 | 83.86 | 83.17 | 85.51 | 87.04 | 87.70 | 88.44 | 89.77 | 89.60 | 91.01 | **91.09** |
| | **20** | 84.16 | 86.84 | 89.41 | 90.56 | 90.22 | 90.69 | 91.02 | **91.13** | 90.87 | 90.68 |
| | 25 | 84.11 | 88.22 | 89.46 | **89.68** | 89.35 | 88.77 | 87.86 | 87.49 | 86.71 | 86.78 |
| | 30 | 85.35 | **88.93** | 89.88 | 89.04 | 88.34 | 87.58 | 87.03 | 85.49 | 85.08 | 85.08 |
| | 35 | 86.17 | **88.80** | 88.67 | 87.67 | 86.63 | 85.84 | 85.26 | 84.87 | 83.95 | 83.06 |
| | 40 | 86.86 | **89.25** | 89.02 | 87.78 | 86.37 | 84.74 | 83.60 | 82.83 | 81.69 | 80.92 |

As can be seen from Table 3, different datasets present different changes in classification accuracy. Compared with the other
two data sets, the scale of Indian Pines is the smallest, so its experimental block window side length values from 3 to 10. With





the increase of the number of iterations, the classification accuracy showed a trend of gradual increase, and the optimal accuracy
was obtained when the side length was 7. When the side length of the block window for Pavia and Salinas datasets is too small,
the classification accuracy will not improve with the increase of iteration times. This indicates that the neighborhood
information set cannot help the sample to distinguish the category at this time.
With the increase of the side length of the block window, the number of iterations to achieve the optimal classification
accuracy is advanced, but the optimal accuracy decreases. The block window is larger, the more noise information will be
introduced, and that will affect the accuracy of sample labeling. Therefore, the block window size of the Indian Pines dataset
is 7 * 7, and the block window sizes of the Pavia and Salinas datasets are 25 * 25 and 20 * 20, respectively.
*5.3.4 Determination of priority classifier*
In fact, the determination of pseudo-tags of samples mainly depends on the determination of classifiers, KNN, SRC, NRS,
MLR are employed to determine the pseudo-tags. The experimental results of single classifier and combination of different
classifiers on different data sets are shown in Table 4 ~ Table 6.

**Table 4** The experimental results of different classifier combinations in Indian Pines data set

| Classifier | Index | 1 | 2 | 3 | 4 | 5 | 6 | 7 | 8 | 9 | 10 |
|---|---|---|---|---|---|---|---|---|---|---|---|
| KNN | NUM | 314 | 673 | 1139 | 1630 | 2178 | 2680 | 3324 | 4063 | 4820 | 5474 |
| | OA(%) | 78.69 | 79.93 | 81.20 | 82.05 | 82.49 | 84.38 | 85.70 | 86.52 | 87.33 | 87.77 |
| SRC | NUM | 311 | 648 | 1068 | 1525 | 2032 | 2606 | 3248 | 3899 | 4668 | **5522** |
| | OA(%) | 78.64 | 80.42 | 80.10 | 81.56 | 82.87 | 84.39 | 85.83 | 87.04 | 87.92 | 88.19 |
| NRS | NUM | 315 | 673 | 1116 | 1597 | 2136 | 2697 | 3265 | 3815 | 4437 | 5016 |
| | OA(%) | 78.82 | 81.02 | 80.96 | 83.71 | 84.94 | 85.57 | 86.87 | 88.01 | 89.25 | **89.42** |
| MLR | NUM | 133 | 295 | 580 | 936 | 1296 | 1790 | 2396 | 3077 | 3858 | 4648 |
| | OA(%) | 77.55 | 79.31 | 82.21 | 83.91 | 83.99 | 85.75 | 87.36 | 87.68 | 88.27 | 88.41 |
| KNN+SRC | NUM | 317 | 706 | 1120 | 1702 | 2294 | 2967 | 3728 | 4494 | 5233 | 5950 |
| | OA(%) | 78.71 | 80.96 | 81.99 | 83.97 | 84.96 | 85.82 | 86.99 | 87.75 | 87.93 | 88.10 |
| KNN+NRS | NUM | 317 | 707 | 1198 | 1691 | 2308 | 2954 | 3625 | 4450 | 5374 | 6305 |
| | OA(%) | 78.78 | 80.27 | 80.73 | 82.51 | 83.82 | 85.45 | 87.56 | 89.06 | 89.95 | 90.19 |
| KNN+MLR | NUM | 318 | 712 | 1206 | 1794 | 2555 | 3398 | 4292 | 5072 | 5783 | 6678 |
| | OA(%) | 78.79 | 80.56 | 82.57 | 86.08 | 87.10 | 87.87 | 88.50 | 88.88 | 89.18 | 89.31 |
| SRC+KNN | NUM | 317 | 706 | 1134 | 1673 | 2223 | 2875 | 3658 | 4317 | 5011 | 5748 |
| | OA(%) | 78.71 | 81.09 | 81.51 | 83.17 | 84.27 | 85.84 | 86.94 | 87.54 | 88.26 | 88.64 |
| SRC+NRS | NUM | 318 | 730 | 1205 | 1778 | 2372 | 3091 | 3969 | 4813 | 5787 | 6641 |
| | OA(%) | 78.81 | 80.79 | 82.37 | 85.03 | 86.81 | 88.45 | 88.93 | 89.86 | 90.49 | 90.78 |
| SRC+MLR | NUM | 315 | 708 | 1153 | 1744 | 2457 | 3322 | 4231 | 5042 | 5800 | 6735 |
| | OA(%) | 78.80 | 80.92 | 82.16 | 85.16 | 86.24 | 87.61 | 88.39 | 88.71 | 89.07 | 89.19 |
| NRS+KNN | NUM | 317 | 707 | 1202 | 1712 | 2385 | 3021 | 3700 | 4538 | 5399 | 6333 |
| | OA(%) | 78.74 | 80.67 | 81.33 | 83.36 | 84.46 | 86.35 | 87.93 | 89.23 | 90.03 | 90.43 |
| NRS+SRC | NUM | 318 | 734 | 1207 | 1728 | 2378 | 3061 | 3959 | 4799 | 5691 | 6739 |
| | OA(%) | 78.77 | 81.01 | 82.56 | 84.52 | 86.37 | 87.43 | 88.95 | 90.11 | 90.61 | 90.90 |
| NRS+MLR | NUM | 318 | 694 | 1148 | 1690 | 2446 | 3194 | 4102 | 4950 | 5768 | 6792 |
| | OA(%) | 78.91 | 81.39 | 81.62 | 85.51 | 87.21 | 89.63 | 90.28 | 90.92 | 91.28 | 91.88 |
| MLR+KNN | NUM | 318 | 677 | 1166 | 1689 | 2429 | 3246 | 4018 | 4737 | 5677 | 6672 |
| | OA(%) | 78.30 | 81.13 | 81.85 | 85.94 | 87.22 | 88.28 | 88.77 | 88.98 | 89.20 | 89.29 |
| MLR+SRC | NUM | 315 | 704 | 1258 | 1741 | 2439 | 3286 | 4097 | 4867 | 5775 | 6760 |
| | OA(%) | 78.31 | 81.81 | 82.55 | 85.09 | 86.86 | 88.30 | 89.01 | 89.44 | 90.21 | 90.71 |
| MLR+NRS | NUM | 318 | 683 | 1154 | 1701 | 2458 | 3301 | 4219 | 5057 | 5997 | **6889** |
| | OA(%) | 78.46 | 81.49 | 82.11 | 86.14 | 87.86 | 89.70 | 90.68 | 91.58 | 92.15 | **92.42** |

From the experimental results on the Indian Pines dataset, it can draw the following conclusions. With the increase of
iterations, the classification accuracy of each sample increased gradually. Compared with the results of the single classifier, the
SRC has the largest number of samples after 10 iterations are finished, but the classification effect is not the best. The classifier



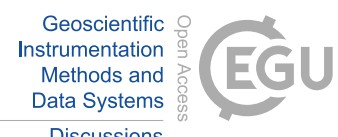
with the best classification effect is NRS. The number of samples and classification accuracy of the combination of two groups
of classifiers are mostly better than that of a single classifier. The experimental results are different for two groups of classifiers
with different priority. The classifier with NRS can achieve more than 90% classification effect after 10 iterations are finished.
The number of combinations with MLR was more than 6600 after 10 iterations are finished, and the best combination was
MLR + NRS after 10 iterations are finished.
**Table 5** The experimental results of different classifiers for Pavia University data

| Classifier | Index | 1 | 2 | 3 | 4 | 5 | 6 | 7 | 8 | 9 | 10 |
|---|---|---|---|---|---|---|---|---|---|---|---|
| KNN | NUM | 236 | 444 | 661 | 965 | 1210 | 1466 | 1812 | 2254 | 2762 | 3251 |
| | OA(%) | 71.19 | 73.36 | 74.70 | 75.69 | 76.10 | 76.16 | 77.28 | 78.48 | 77.60 | 77.69 |
| SRC | NUM | 225 | 477 | 733 | 1080 | 1492 | 2047 | 2793 | 3654 | 4605 | 5631 |
| | OA(%) | 72.21 | 72.19 | 76.97 | 79.12 | 80.08 | 81.77 | 83.09 | 84.07 | 84.95 | 85.27 |
| NRS | NUM | 225 | 438 | 763 | 1007 | 1256 | 1487 | 1726 | 1893 | 1991 | 2179 |
| | OA(%) | 71.22 | 73.74 | 75.14 | 76.59 | 76.15 | 76.47 | 76.41 | 76.20 | 75.68 | 75.40 |
| MLR | NUM | 100 | 244 | 396 | 605 | 829 | 1035 | 1304 | 1587 | 1872 | 2183 |
| | OA(%) | 72.36 | 76.37 | 77.18 | 79.04 | 79.40 | 81.16 | 82.21 | 82.18 | 83.36 | **85.85** |
| KNN+SRC | NUM | 248 | 473 | 787 | 1118 | 1509 | 1996 | 2552 | 3111 | 3975 | 4837 |
| | OA(%) | 71.48 | 72.82 | 74.61 | 76.59 | 78.38 | 80.08 | 80.82 | 81.62 | 82.40 | 83.99 |
| KNN+NRS | NUM | 240 | 491 | 775 | 1067 | 1382 | 1850 | 2403 | 3040 | 3794 | 4415 |
| | OA(%) | 71.11 | 73.83 | 77.44 | 78.83 | 79.50 | 79.58 | 79.39 | 80.11 | 80.74 | 81.48 |
| KNN+MLR | NUM | 244 | 515 | 822 | 1176 | 1616 | 2162 | 2905 | 3698 | 4745 | 5682 |
| | OA(%) | 71.37 | 75.03 | 77.84 | 78.08 | 79.73 | 79.59 | 81.30 | 83.88 | 84.90 | 85.08 |
| SRC+KNN | NUM | 248 | 476 | 795 | 1215 | 1725 | 2298 | 3055 | 3967 | 4977 | 5913 |
| | OA(%) | 71.38 | 72.54 | 75.26 | 78.09 | 79.19 | 80.21 | 81.88 | 83.31 | 83.35 | 83.51 |
| SRC+NRS | NUM | 242 | 511 | 867 | 1385 | 1889 | 2513 | 3201 | 3988 | 4910 | 5794 |
| | OA(%) | 71.40 | 74.12 | 77.56 | 80.80 | 82.19 | 82.55 | 83.02 | 83.80 | 84.34 | 85.09 |
| SRC+MLR | NUM | 236 | 507 | 841 | 1289 | 1731 | 2261 | 3016 | 3928 | 4939 | 6015 |
| | OA(%) | 71.52 | 73.62 | 76.47 | 78.45 | 79.51 | 81.54 | 83.10 | 84.37 | 84.68 | 84.66 |
| NRS+KNN | NUM | 240 | 486 | 803 | 1119 | 1541 | 1992 | 2557 | 3190 | 3939 | 4611 |
| | OA(%) | 71.05 | 74.37 | 76.57 | 77.77 | 78.06 | 77.21 | 76.75 | 77.35 | 78.88 | 79.28 |
| NRS+SRC | NUM | 242 | 501 | 803 | 1296 | 1792 | 2433 | 3089 | 3839 | 4776 | 5798 |
| | OA(%) | 71.44 | 74.50 | 76.72 | 79.93 | 81.44 | 81.90 | 82.85 | 83.81 | 85.00 | 85.48 |
| NRS+MLR | NUM | 234 | 517 | 828 | 1237 | 1796 | 2446 | 3354 | 4220 | 5061 | 5857 |
| | OA(%) | 71.49 | 75.47 | 77.89 | 80.86 | 83.68 | 84.43 | 84.93 | 85.12 | 85.57 | 85.46 |
| MLR+KNN | NUM | 244 | 486 | 746 | 1170 | 1658 | 2300 | 3205 | 4208 | 5336 | **6514** |
| | OA(%) | 71.47 | 75.30 | 76.67 | 78.76 | 80.35 | 82.97 | 85.05 | 86.05 | 86.88 | 87.02 |
| MLR+SRC | NUM | 236 | 504 | 903 | 1310 | 1708 | 2207 | 3009 | 4024 | 5098 | 6234 |
| | OA(%) | 71.71 | 74.01 | 79.30 | 79.81 | 80.40 | 83.03 | 86.27 | 86.93 | 87.97 | **88.53** |
| MLR+NRS | NUM | 234 | 524 | 787 | 1205 | 1738 | 2374 | 3116 | 3953 | 4754 | 5586 |
| | OA(%) | 71.63 | 75.94 | 79.83 | 80.66 | 82.75 | 84.64 | 85.98 | 86.19 | 86.37 | 86.87 |

As can be seen in Table 5, compared with the experimental results by single classifier, the number of labeled samples with
SRC is the largest after 10 iterations are finished, which is higher than the other three methods. However, the MLR obtained
best classification results. After 10 iterations are finished, the number of labeled samples of the two classifiers is more than that
of the single classifier. With KNN, SRC and NRS as the first priority classifiers, the classification results of the sample set after
10 iterations are not as good as those obtained by using MLR. The combination of MLR as the first priority classifier has better
classification effect than single MLR after 10 iterations are finished.
**Table 6** The experimental results of different classifiers for Salinas Scene data

| Classifier | Index | 1 | 2 | 3 | 4 | 5 | 6 | 7 | 8 | 9 | 10 |
|---|---|---|---|---|---|---|---|---|---|---|---|
| KNN | NUM | 133 | 251 | 443 | 684 | 963 | 1304 | 1672 | 2047 | 2425 | 2857 |
| | OA(%) | 83.38 | 86.46 | 86.46 | 86.65 | 87.26 | 87.31 | 87.79 | 87.92 | 87.66 | 87.15 |

| | | | | | | | | | | | |
|---|---|---|---|---|---|---|---|---|---|---|---|
| SRC | NUM | 144 | 275 | 441 | 666 | 937 | 1255 | 1606 | 1968 | 2391 | 2811 |
| | OA(%) | 83.10 | 83.56 | 85.11 | 85.83 | 85.81 | 86.63 | 86.50 | 86.64 | 86.95 | 86.96 |
| NRS | NUM | 148 | 271 | 423 | 668 | 952 | 1263 | 1569 | 1940 | 2351 | 2799 |
| | OA(%) | 84.04 | 86.06 | 87.62 | 87.63 | 87.10 | 87.48 | 88.44 | 88.43 | 88.32 | 87.92 |
| MLR | NUM | 102 | 177 | 302 | 451 | 621 | 867 | 1176 | 1518 | 1848 | 2217 |
| | OA(%) | 82.88 | 85.41 | 87.11 | 88.36 | 88.73 | 90.68 | 91.52 | 92.20 | 91.70 | 92.02 |
| KNN+SRC | NUM | 146 | 294 | 479 | 694 | 976 | 1330 | 1691 | 2106 | 2520 | 2985 |
| | OA(%) | 83.60 | 84.65 | 85.38 | 86.76 | 87.39 | 87.39 | 88.00 | 88.11 | 87.53 | 87.23 |
| KNN+NRS | NUM | 150 | 297 | 500 | 761 | 1108 | 1466 | 1891 | 2354 | 2834 | 3316 |
| | OA(%) | 83.53 | 86.43 | 87.14 | 87.49 | 87.64 | 87.37 | 88.06 | 88.03 | 87.95 | 88.03 |
| KNN+MLR | NUM | 143 | 285 | 508 | 768 | 1132 | 1546 | 2026 | 2526 | 3049 | 3590 |
| | OA(%) | 83.38 | 85.50 | 86.34 | 88.35 | 88.80 | 90.07 | 90.24 | 90.08 | 89.86 | 89.79 |
| SRC+KNN | NUM | 146 | 287 | 472 | 686 | 957 | 1281 | 1673 | 2061 | 2485 | 2892 |
| | OA(%) | 83.25 | 84.70 | 85.79 | 85.88 | 86.70 | 86.92 | 86.89 | 86.93 | 86.85 | 86.90 |
| SRC+NRS | NUM | 150 | 280 | 493 | 755 | 1017 | 1328 | 1708 | 2114 | 2539 | 2982 |
| | OA(%) | 83.07 | 85.55 | 86.28 | 85.47 | 86.66 | 86.97 | 87.52 | 87.05 | 87.19 | 87.16 |
| SRC+MLR | NUM | 150 | 271 | 483 | 720 | 1108 | 1499 | 1958 | 2451 | 2969 | 3487 |
| | OA(%) | 83.15 | 84.27 | 87.25 | 87.88 | 88.88 | 89.29 | 89.36 | 89.85 | 89.70 | 89.85 |
| NRS+KNN | NUM | 150 | 298 | 519 | 814 | 1148 | 1556 | 2037 | 2538 | 3027 | 3522 |
| | OA(%) | 84.01 | 87.12 | 87.79 | 87.30 | 87.54 | 88.38 | 88.09 | 88.25 | 87.95 | 87.88 |
| NRS+SRC | NUM | 150 | 284 | 488 | 803 | 1158 | 1539 | 1988 | 2482 | 2955 | 3423 |
| | OA(%) | 83.91 | 87.23 | 86.98 | 87.73 | 88.30 | 88.97 | 89.57 | 89.57 | 89.60 | 89.43 |
| NRS+MLR | NUM | 153 | 293 | 509 | 762 | 1104 | 1509 | 1940 | 2441 | 2934 | 3452 |
| | OA(%) | 83.87 | 85.82 | 88.25 | 89.93 | 90.40 | 90.51 | 90.75 | 90.48 | 90.12 | 89.61 |
| MLR+KNN | NUM | 143 | 299 | 521 | 825 | 1187 | 1602 | 2046 | 2514 | 2993 | 3407 |
| | OA(%) | 82.80 | 85.42 | 87.67 | 88.79 | 89.68 | 90.06 | 90.54 | 90.92 | 91.27 | 91.32 |
| MLR+SRC | NUM | 150 | 292 | 521 | 799 | 1123 | 1537 | 2007 | 2448 | 2929 | 3367 |
| | OA(%) | 82.91 | 85.45 | 88.89 | 90.12 | 90.21 | 90.93 | 90.99 | 91.83 | 92.08 | **92.64** |
| MLR+NRS | NUM | 153 | 315 | 564 | 841 | 1197 | 1605 | 2064 | 2561 | 3060 | 3500 |
| | OA(%) | 82.81 | 84.87 | 88.16 | 89.34 | 89.69 | 90.03 | 90.42 | 90.52 | 91.19 | 91.43 |

For Salinas Scene data, the number of labeled samples by MLR after 10 iterations is the smallest, but the classification accuracy of the labeled samples is the highest. After 10 iterations are finished, the number of iterations of the two classifiers is also higher than that of the single classifier. However, from the perspective of the performance of labeled samples in classification, MLR+SRC has higher classification results than MLR, which indicates that the addition of classifiers does not necessarily improve the classification accuracy, and experiments and analysis are needed for different data sets.

From the three experiments, it can be seen that the method with the largest number of labeled samples does not necessarily achieve the best classification results. The labeled samples are needed to improve the classification accuracy of the classifier, so the obtained labeled samples after 10 iterations are taken as the evaluation criteria. The Indian Pines data set uses a combination of classifiers MLR + NRS. The Pavia University and Salinas Scene data sets use MLR + SRC.

*5.4. Experimental results and analysis*

Based on the above analysis, the related parameters are shown in Table 7.

Table 7 The parameter settings

| Data set | Indian Pines | Pavia University | Salinas Scene |
|---|---|---|---|
| Selection policy | BT | BT | BT |
| Number of selections | 1600 | 1400 | 600 |
| Window size | 7*7 | 25*25 | 20*20 |
| Combination of classifiers | MLR+NRS | MLR+SRC | MLR+SRC |
| Number of labeled samples | 6889 | 6234 | 3367 |



Firstly, the local binary pattern is used to extract the features of spatial texture information of hyperspectral remote sensing
images. Secondly, the sample labeling method based on neighborhood information and priority classifier is proposed to obtain
the learned pseudo-labeled samples. Then the SRC classifier is trained with the labeled samples, and the test samples are
predicted. The obtained classification results are compared with those obtained by the SRC classifier on the initial training data,
and the classification results of training models with different training data are shown in Table 8.

**Table 8** The classification results of training models with different training data

| Training samples | Index | Initial samples | Labeling samples |
|---|---|---|---|
| Indian Pines | AA | 67.93% | 84.70% |
| | OA | 77.38% | 92.42% |
| | KAPPA | 0.746 | 0.914 |
| Pavia University | AA | 60.53% | 81.87% |
| | OA | 69.00% | 88.53% |
| | KAPPA | 0.609 | 0.848 |
| Salinas Scene | AA | 82.59% | 87.76% |
| | OA | 84.00% | 92.64% |
| | KAPPA | 0.823 | 0.918 |

The classification visualizations of the classification model for the initial samples and labeled samples are shown in Figure
7 and Figure 8.

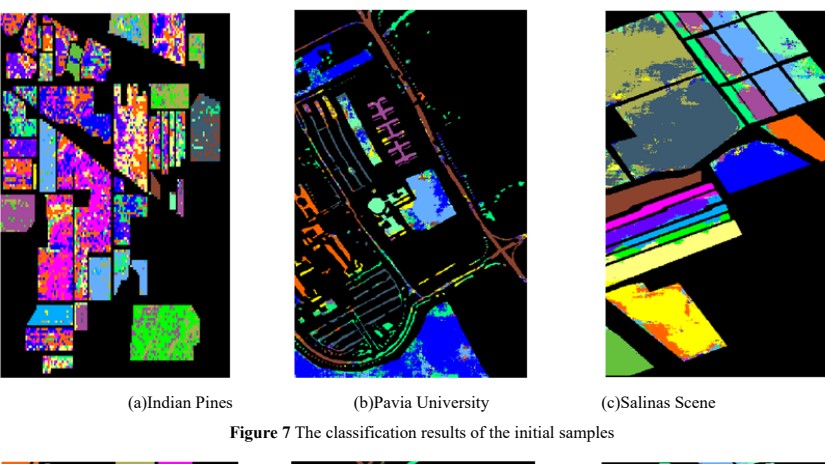

10               (a)Indian Pines                    (b)Pavia University              (c)Salinas Scene
11                            **Figure 7** The classification results of the initial samples

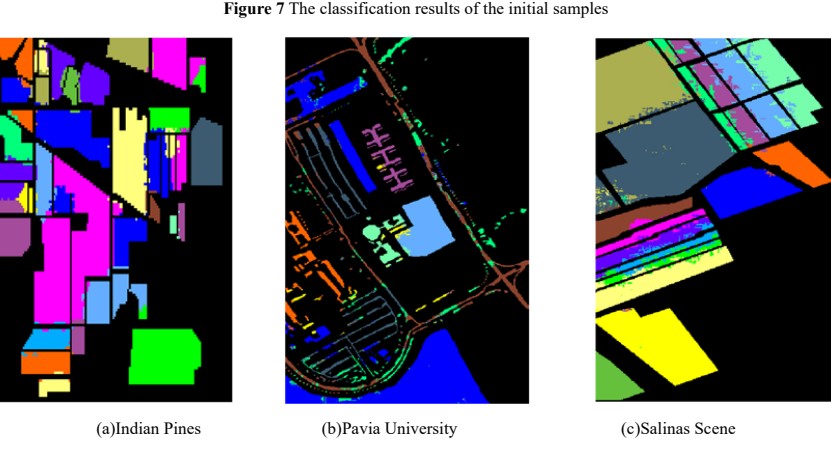

14               (a)Indian Pines                    (b)Pavia University              (c)Salinas Scene
15                            **Figure 8** The classification results of the labeling samples





By comparing with the results of the experiments, it is not difficult to find that the classification results of the classifier
trained with expanded samples on the three sets of data are better than those of the classifier trained with initial samples.
Moreover, from the classification visualization, it can see that the obtained classification results by the classifier and the labeled
samples is smoother and has fewer discrete points, which indicates that the generalization ability of the classifier is improved
by labeling the samples.

## 6. Conclusion

For the difficulties of hyperspectral image processing and analysis, a hyperspectral remote sensing image classification method based on texture features and semi-supervised learning is implemented by introducing local binary model, sparse representation and mixed logistic regression model. The local binary pattern is employed to deal with the hyperspectral data and extract the texture features of the hyperspectral remote sensing image. A sample labeling method based on neighborhood information and priority classifier is proposed to obtain the learned pseudo-labeled samples. The problem of limited labeled samples of hyperspectral images is solved. The data of Indian Pines, Salinas scene and Pavia University are selected in here. The experiment results of the BT method are obviously better than those of other methods. The block window of Indian Pines dataset is 7*7. The block windows of Pavia University and Salinas scene are 25 * 25 and 20 * 20, respectively. The combination of MLR and SRC can get better classification results. The obtained classification results by the classifier and the labeled samples are smoother and has fewer discrete points, which indicates that the generalization ability of the classifier is improved by labeling the samples from the classification visualization.

## Author Contributions

Conceptualization, Ansheng Ye and Xiangbing Zhou; Methodology, Ansheng Ye and Yu Gong; Software, Yu Gong.; Validation, Fang Miao and Yu Gong; Resources, Fang Miao; Writing—original draft preparation, Ansheng Ye and Yu Gong; Writing—review and editing, Xiangbing Zhou and Huimin Zhao; Visualization, Fang Miao; Project administration, Huimin Zhao; Funding acquisition, Xiangbing Zhou. All authors have read and agreed to the published version of the manuscript.

## Acknowledgments

This research was funded by the Sichuan Science and Technology Program, grant number 2019ZYZF0169, 2019YFG0307, 2021YFS0407; the A Ba Achievements Transformation Program, grant number R21CGZH0001; the Chengdu Science and technology planning project, grant number 2021-YF05-00933-SN.

## Data Availability Statement

Not applicable

## Conflicts of Interest

The authors declare no conflict of interest.

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
