# Peer review of "Sample labeling and classification method of hyperspectral remote sensing images based on texture features and semi-supervised learning"

_Geoscientific Instrumentation, Methods and Data Systems, 2022_

## Author Comment (AC1)

**Dear reviewer RC1:**

Thank you very much for the insightful comments. Thank you for giving us a choice to correct the shortcoming of our manuscript. We already carefully read your comments and revised the manuscript according to your suggestions. We hope that this revision will make our manuscript meet the publisher. The responses to the comments point by point are listed below. Please feel free to contact us with any questions. If the revised manuscript maybe exists the shortcomings, please tell us. We will try our best to continue to revise our manuscript in order to improve our manuscript. Really thank your insightful comments and help again!

Yours sincerely,

Regards,

Xianging Zhou

**Reviewer #1:**

The results look encouraging and motivating. But some contents need be revised in order to meet the requirements of publish.

(1)The abstract should be improved. Your point is your own work that should be further highlighted.

(2)The parameters in expressions are given and explained.

(3) The method in the context of the proposed work should be written in detail.

(4) The values of parameters could be a complicated problem itself, how the authors give the values of parameters in the used methods.

(5) The literature review is poor in this paper. You must review all significant similar works that have been done. I hope that the authors can add some new references in order to improve the reviews and the connection with the literatures.

(6) The main contributions of this paper should be further summarized and clearly demonstrated. This reviewer suggests the authors exactly mention what is new compared with existing.

(7) The conclusion and motivation of the work should be added in a clearer way.

(8) There are some grammatical errors seen in the paper. Check carefully for a few clerical errors and formatting issues.

**COMMENT 1:** The abstract should be improved. Your point is your own work that should be further highlighted.

**RESPONSE:** Thank you very much for the insightful comments. According to expert advice, we have substantially modified our manuscript in order to clearly describe the abstract and highlight the contributions. In this study, local binary pattern (LBP), sparse representation and mixed logistic regression model are introduced to propose a sample labeling method based on neighborhood information and priority classifier discrimination. Then, a hyperspectral remote sensing image classification method based on texture features and semi-supervised learning is implemented. The data of Indian Pines, Salinas scene and Pavia University are selected to verify the validity of the proposed method. The experiment

results show that the proposed classification method obtains higher classification accuracy and shows stronger timeliness and generalization ability. Please read our revised manuscript, thanks!

*Abstract: Hyperspectral images contain abundant spectral and spatial information of the surface of earth, which increase the difficulties of data processing and analysis, and sample labeling. In this paper, local binary pattern (LBP), sparse representation and mixed logistic regression model are introduced to propose a sample labeling method based on neighborhood information and priority classifier discrimination. Then, a hyperspectral remote sensing image classification method based on texture features and semi-supervised learning is implemented. The LBP is employed to extract features of spatial texture information from remote sensing images and enrich the feature information of samples. The multivariate logistic regression model is used to select the unlabeled samples with the largest amount of information, and the unlabeled samples with neighborhood information and priority classifier tags are selected to obtain the pseudo-labeled samples after learning. By making full use of the advantages of sparse representation and mixed logistic regression model, a new classification model based on semi-supervised learning is constructed to effectively achieve accurate classification of hyperspectral images. The data of Indian Pines, Salinas scene and Pavia University are selected to verify the validity of the proposed method. The experiment results show that the proposed classification method obtains higher classification accuracy and shows stronger timeliness and generalization ability.*

**COMMENT 2:** The parameters in expressions are given and explained.

**RESPONSE:** Thank you very much for the insightful comments. According to expert advice, we have substantially modified our manuscript in order to explain the parameters in expressions, such as expression (1), expression(4), ….. Please read our revised manuscript, thanks!

**COMMENT 3:** The method in the context of the proposed work should be written in detail.

**RESPONSE:** Thank you very much for the insightful comments. According to expert advice, we have substantially modified our manuscript in order to write the method in the context of the proposed work in detail. Please read our revised manuscript, thanks!

[revised manuscript text omitted]

**COMMENT 4:** The values of parameters could be a complicated problem itself, how the authors give the values of parameters in the used methods.

**RESPONSE:** Thank you very much for the insightful comments. According to expert advice, we have substantially modified our manuscript in order to give the values of parameters in the used methods. In our study, a large number of alternative values are tested, and some classical values are selected from literatures, then these parameter values are experimentally modified until the most reasonable parameter values are determined. These selected parameter values have obtained the optimal solution, so that they can accurately and efficiently verify the effectiveness. Please read our revised manuscript, thanks!

**COMMENT 5:** The literature review is poor in this paper. You must review all significant similar works that have been done. I hope that the authors can add some new references in order to improve the reviews and the connection with the literatures.

**RESPONSE:** Thank you very much for the insightful comments. According to expert advice, we have substantially modified our manuscript in order to review all significant similar works and add some new references in order to improve the reviews and the connection with the literatures. Please read our revised manuscript, thanks!

Camps-Valls G, Bandos T, Zhou D. (2007). Semi-supervised graph-based hyperspectral image classification. IEEE Transactions on Geoscience and Remote Sensing, 45(10):3044-3054.

Chang C, Kuo Y, Chen S, Liang C, Ma KY, Hu PF. (2021). Self-Mutual information-based band selection for hyperspectral image classification. IEEE Trans Geoscience and Remote Sensing, 59 (7): 5979-5997.

Chen C, Ma Y, Ren G. (2020) .Hyperspectral classification using deep belief networks based on conjugate gradient update and pixel-centric spectral block features. IEEE Journal of Selected Topics in Applied Earth Observations and Remote Sensing, 13:4060-4069.

Chen GY. (2021). Multiscale filter-based hyperspectral image classification with PCA and SVM. Journal of Electrical Engineering, 72(1):pp. 40-45

Chen, HY, Fang M, Xu S. (2020). Hyperspectral remote sensing image classification with CNN based on quantum genetic-optimized sparse representation. IEEE Access, 8: 99900-99909.

Chen H, Miao F, Chen Y. (2021). A hyperspectral image classification method using multifeature vectors and optimized KELM. IEEE Journal of Selected Topics in Applied Earth Observations and Remote Sensing, 14: 2781-2795.

Chen Y, Nasser MN, Tran TD. (2011). Hyperspectral image classification using dictionary-based sparse representation. IEEE Transactions on Geoscience and Remote Sensing, 49(10):3973-3985.

Chen Y, Nasser MN, Tran TD. (2013). Hyperspectral image classification via kernel sparse representation. IEEE Transactions on Geoscience and Remote Sensing, 51(1):217–231.

Cui M, Prasad S. (2013).Multiscale sparse representation classification for robust hyperspectral image analysis. IEEE Global Conference on Signal and Information Processing, 969-972.

Deng W, Zhang L, Zhou X, et al. (2022). Multi-strategy particle swarm and ant colony hybrid optimization for airport taxiway planning problem. Information Sciences, 612: 576-593.

Duan Z, Song P, Yang C, et al. (2022). The impact of hyperglycaemic crisis episodes on long-term outcomes for inpatients presenting with acute organ injury: A prospective, multicentre follow-up study. Frontiers in Endocrinology, Doi: 10.3389/fendo.2022.1057089

Dou Z, Gao K, Zhang X, Wang H, Han L. (2020). Band selection of hyperspectral images using attention-based autoencoders. IEEE Geoscience and Remote Sensing Letters, 18 (1): 147-151.

Dumke I, Ludvigsen M, Ellefmo SL, Søreide F, Johnsen G, Murton B. (2019).  Underwater hyperspectral imaging using a stationary platform in the transatlantic geotraverse hydrothermal field. IEEE Transactions on Geoscience and Remote Sensing, 57 (5): 2947-2962.

Huang C, Zhou X, Ran X, et al.(2023). Co-evolutionary competitive swarm optimizer with three-phase for large-scale complex optimization problem. Information Sciences, 619:2-18.

Huang C, Zhou X, Ran X, et al.(2023). Adaptive cylinder vector particle swarm optimization with differential evolution for UAV path planning . Engineering Applications of Artificial Intelligence,121:105942.

Huang W, Huang Y, Wang H, Liu Y, Shim HJ. (2020). Local binary patterns and superpixel-based multiple kernels for hyperspectral image classification. IEEE Journal of Selected Topics in Applied Earth Observations and Remote Sensing, 13: 4550-4563.

Hu S, Xu C, Peng J, Yan X, Long T. (2019).Weighted Kernel joint sparse representation for hyperspectral image classification. IET Image Processing, 13(2):254-260.

Jiang X, Liu W, Zhang Y, Liu J, Li S, Lin J. (2020). Spectral-spatial hyperspectral image classification using dual-channel capsule networks. IEEE Geoscience and Remote Sensing Letters, 18 (6): 1094-1098.

*Liu ZX, Ma L, Du Q. (2021).Class-wise distribution adaptation for unsupervised classification of hyperspectral remote sensing images. IEEE Transactions on Geoscience and Remote Sensing, 59(1): 508-521*

*Melgani F, Bruzzone L. (2004). Classification of hyperspectral remote sensing images with support vector machines. 2004. IEEE Transactions on Geoscience and Remote Sensing, 42(8):1778-1790.*

*Ojala T, Harwood I. (1996).A comparative study of texture measures with classification based on feature distributions. Pattern Recognition29(1):51-59.*

*Ratle, Terrettaz-Zufferey, Kanevski, et al. (206). Learning manifolds in forensic data. international conference on artificial neural networks. Springer, Berlin, Heidelberg.*

*Samiappan S, Moorhead R J. (2015). Semi-supervised co-training and active learning framework for hyperspectral image classification.2015 IEEE International Geoscience and Remote Sensing Symposium(IGARSS), IEEE:401-404.*

*Seifi M, Ghassemian H. (2017). A probabilistic SVM approach for hyperspectral image classification using spectral and texture features. International Journal of Remote Sensing, 38 (15): 4265-4284.*

*Shang X, Song M, Chang CI. (2020). An iterative random training sample selection approach to constrained energy minimization for hyperspectral image classification. IEEE Geoscience and Remote Sensing Letters, 18 (9): 1625-1629.*

*Shi C, Pun CM. (2019). Multiscale superpixel-based hyperspectral image classification using recurrent neural networks with stacked autoencoders. IEEE Transactions on Multimedia, 22 (2): 487-501.*

*Song Y, Cai X, Zhou X, et al. (2022). Dynamic hybrid mechanism-based differential evolution algorithm and its application. Expert Systems with Applications, 213: 118834.*

*Song Y, Zhao G, Zhang B, et al.(2023). An enhanced distributed differential evolution algorithm for portfolio optimization problems. Engineering Applications of Artificial Intelligence,121:106004*

*Tan K, Li E, Qian D, et al. (2014). An efficient semi-supervised classification approach for hyperspectral imagery. ISPRS Journal of Photogrammetry & Remote Sensing, 97:36-45.*

*Tang YY, Yuan H, Li L. (2014).Manifold-based sparse representation for hyperspectral image classification. IEEE Transactions on Geoscience and Remote Sensing, 52(12):7606-7618.*

*Wang C, Wang H, Hu B, Jia W, Xu J, Li X. (2016). A novel spatial-spectral sparse representation for hyperspectral image classification based on neighborhood segmentation. Spectroscopy and Spectral Analysis, 36(9):2919-2924.*

*Wang HR, Celik T. (2018). Sparse representation-based hyperspectral image classification. Signal Image and Video Processing, 12(5):1009-1017.*

*Wang QY, Zhang Q, Zhang JP, Kang SQ, Wang YJ. (2022). Graph-based semisupervised learning with weighted features for hyperspectral remote sensing image classification. IEEE Journal of Selected Topics in Applied Earth Observations and Remote Sensing, 15: 6356-6370*

*Xu J.; Zhao Y.; Chen H.; Deng W.(2023). ABC-GSPBFT: PBFT with grouping score mechanism and optimized consensus process for flight operation data-sharing. Information Sciences, 624:110-127.*

*Xue ZH, Du PJ, Li J, Su HJ. (2017). Sparse graph regularization for hyperspectral remote sensing image classification. IEEE Transactions on Geoscience and Remote Sensing, 55(4): 2351-2366*

*Yang C, Liu S C, Bruzzone L, et al. (2012). A semisupervised feature metric-based band selection method for hyperspectral image classification. Hyperspectral Image and Signal Processing (WHISPERS), 2012 4th Workshop on. IEEE.*

*Yang M, Li CH, Guan J, Yan XS. (2018). A supervised-learning p-norm distance metric for hyperspectral remote sensing image classification. IEEE Geoscience and Remote Sensing Letters, 15(9): 1432-1436*

*Yang X, Cao W, Lu Y, et al. (2022). Hyperspectral image transformer classification networks. IEEE Transactions on Geoscience and Remote Sensing 60: 1- 15. doi:10.1109/TGRS.2022.3171551*

*Ye X, Ma J, Xiong H. (2021). Local affine preservation with motion consistency for feature matching of remote sensing images. IEEE Transactions on Geoscience and Remote Sensing, 60: 1-12.*

*Yin J, Qi C, Chen Q, Qu J. (2021). Spatial-spectral network for hyperspectral image classification: A 3-D CNN and Bi-LSTM framework. Remote Sensing, 13 (12), 2353.*

*Yu C, Liu C, Yu H, Song M, Chang CI.(2021). Unsupervised domain adaptation with dense-based compaction for hyperspectral imagery..IEEE Journal of Selected Topics in Applied Earth Observations and Remote Sensing 14: 12287–12299.*

*Yu C, Zhou S, Song M, Chang CI. (2021). Semisupervised hyperspectral band selection based on dual-constrained low-rank representation. IEEE Geoscience and Remote Sensing Letters, 19: 1-5.*

*Zhang CJ, Li GD, Du SH. (2019). Multi-scale dense networks for hyperspectral remote sensing image classification. IEEE Transactions on Geoscience and Remote Sensing, 57(11): 9201-9222*

*Zhang J, Meng Z, Zhao F, Liu H, (2022). Chang Z. Convolution transformer mixer for hyperspectral image classification. IEEE Geoscience and Remote Sensing Letters, doi:10.1109/LGRS.2022.3208935.*

*Zhao H, Wang C, Chen H, Chen T, Deng W.(2023). A hybrid classification method with dual-channel CNN and KELM for hyperspectral*

*remote sensing images, International Journal of Remote Sensing, 44(1):289-310*

*Zhang X, Wang H, Du C, et al.(2022). Custom-molded offloading footwear effectively prevents recurrence and amputation, and lowers mortality rates in high-risk diabetic foot patients: a multicenter, prospective observational study. Diabetes, Metabolic Syndrome and Obesity: Targets and Therapy, 15: 103-109*

*Zhao XD, Zhang MM, Tao R, Li W, Liao WZ, Tian LF, Philips W. (2022). Fractional Fourier image transformer for multimodal remote sensing data classification. IEEE Transactions on Neural Networks and Learning Systems, doi: 10.1109/TNNLS.2022.3189994*

*Zhang Z, Crawford M. (2016). Semi-supervised multi-metric active learning for classification of hyperspectral images.2016 IEEE International Geoscience and Remote Sensing Symposium(IGARSS), IEEE:1843-1847.*

*Zhao X, Zhang M, Tao R, et al.(2022). Fractional Fourier image transformer for multimodal remote sensing data classification. IEEE Transactions on Neural Networks and Learning Systems 1–13. doi:10.1109/TNNLS.2022.3189994.*

*Zhong K, Zhou G, Deng W, et al. (2021). MOMPA: Multi-objective marine predator algorithm. Computer Methods in Applied Mechanics and Engineering, 385:114029.*

*Zhou S, Xue Z, Du P. (2019). Semisupervised stacked autoencoder with cotraining for hyperspectral image classification. IEEE Transactions on Geoscience and Remote Sensing 57 (6): 3813–3826.*

**COMMENT 6:** The main contributions of this paper should be further summarized and clearly demonstrated. This reviewer suggests the authors exactly mention what is new compared with existing.

**RESPONSE:** Thank you very much for the insightful comments. According to expert advice, we have substantially modified our manuscript in order to further summarize and clearly demonstrate the main contributions of this paper. In addition, we exactly mention what is new compared with existing. In this study, a new sample labeling method based on neighborhood information and priority classifier discrimination is developed to implement a new hyperspectral remote sensing image classification method based on texture features and semi-supervised learning. Please read our revised manuscript, thanks!

**COMMENT 7:** The conclusion and motivation of the work should be added in a clearer way.

**RESPONSE:** Thank you very much for the insightful comments. According to expert advice, we have substantially modified our manuscript in order to add the conclusion and motivation of the work. Please read our revised manuscript, thanks!

*6. Conclusion*

*For the difficulties of hyperspectral image processing and analysis, a new sample labeling method based on neighborhood information and priority classifier discrimination is developed to implement a new hyperspectral remote sensing image classification method based on texture features and semi-supervised learning by introducing local binary*

*model, sparse representation and mixed logistic regression model. The local binary pattern is employed to deal with the hyperspectral data and extract the texture features of the hyperspectral remote sensing image. The multivariate logistic regression model is used to select the unlabeled samples with the largest amount of information, and the unlabeled samples with neighborhood information and priority classifier tags are selected to obtain the pseudo-labeled samples after learning. The problem of limited labeled samples of hyperspectral images is solved. The data of Indian Pines, Salinas scene and Pavia University are selected in here. The experiment results of the BT method are obviously better than those of other methods. The block window of Indian Pines dataset is 7\*7. The block windows of Pavia University and Salinas scene are 25 \* 25 and 20 \* 20, respectively. The combination of MLR and SRC can get better classification results. The obtained classification results by the classifier and the labeled samples are smoother and has fewer discrete points, which indicates that the generalization ability of the classifier is improved by labeling the samples from the classification visualization. For Indian Pines data, the classification results of AA, OA and KAPPA are 84.7%, 94.42% and 0.914, respectively. For Pavia University data, the classification results of AA, OA and KAPPA are 81.87%, 88.53% and 0.848, respectively. For Salinas Scene data, the classification results of AA, OA and KAPPA are 87.76%, 92.64% and 0.918, respectively. Therefore, the classification method obtains the higher classification accuracy.*
*However, the proposed classification method has the more computing time, so the next step should be more in-depth research to reduce the time complexity.*

**COMMENT 8:** There are some grammatical errors seen in the paper. Check carefully for a few clerical errors and formatting issues.

**RESPONSE:** Thank you very much for the insightful comments. According to expert advice, we have substantially modified our manuscript in order to eliminate a number of grammatical errors and spelling errors. In addition, we have invited an English teacher whose native language is English to check the manuscript carefully in order to improve the written English level and avoid solecism and spelling mistakes. Let the revised manuscript be more readable. Please read our revised manuscript, thanks!

**And so on, please read our revised manuscript. We thank the comments and the opportunity for us to improve our manuscript. As much as possible, the questions were taken into account during the preparation of the revised manuscript. We hope that the manuscript is now suitable for publication.**

---

## Author Comment (AC2)

**Dear reviewer RC2:**

Thank you very much for the insightful comments. Thank you for giving us a choice to correct the shortcoming of our manuscript. We already carefully read your comments and revised the manuscript according to your suggestions. We hope that this revision will make our manuscript meet the publisher. The responses to the comments point by point are listed below. Please feel free to contact us with any questions. If the revised manuscript maybe exists the shortcomings, please tell us. We will try our best to continue to revise our manuscript in order to improve our manuscript. Really thank your insightful comments and help again!

Yours sincerely,

Regards,

Xianging Zhou

**Reviewer #1:**

The authors proposed a hyperspectral remote sensing image classification method based on texture features and semi-supervised learning. The LBP is employed to extract features of spatial texture information from remote sensing images and enrich the feature information of samples. Then the multivariate logistic regression model is used to select the unlabeled samples with the largest amount of information, and the unlabeled samples with neighborhood information and priority classifier tags are selected to obtain the pseudo-labeled samples after learning. By making full use of the advantages of sparse representation and mixed logistic regression model, a new hyperspectral remote sensing image classification model based on semi-supervised learning is constructed to effectively achieve accurate classification of hyperspectral images. However, it requires further improvements.

(1) In the abstract section, I would suggest that the author should provide to the point and quantitative advantages of the proposed method.

(2) In the introduction, the authors should clearly indicate the contributions and innovations of this paper.

(3) All acronyms and variables in equations must be defined in the article.

(4) In Section 3.2, how to realize the sample labeling by using neighborhood information and priority classifier?

(5) Figure 2 and Figure3 are not clear, please provide some clear figures.

(6) Why did you use the selected evaluation criteria? What are their advantages?

(7) There are some grammatical mistakes and typo errors. Please proof read from native speaker.

(8) Please add what the next work of this article is.

(9) Some new references should be added to improve the reviews the literatures.

**COMMENT 1:** In the abstract section, I would suggest that the author should provide to the point and quantitative advantages of the proposed method.

**RESPONSE:** Thank you very much for the insightful comments. According to expert advice, we have substantially modified our manuscript in order to provide to the point and quantitative advantages of the proposed method. In this study, local binary pattern (LBP), sparse representation and mixed logistic

regression model are introduced to propose a sample labeling method based on neighborhood information and priority classifier discrimination. Then, a hyperspectral remote sensing image classification method based on texture features and semi-supervised learning is implemented. The experiment results show that the proposed classification method obtains higher classification accuracy and shows stronger timeliness and generalization ability. Please read our revised manuscript, thanks!

*Abstract:Hyperspectral images contain abundant spectral and spatial information of the surface of earth, which increase the difficulties of data processing and analysis, and sample labeling. In this paper, local binary pattern (LBP), sparse representation and mixed logistic regression model are introduced to propose a sample labeling method based on neighborhood information and priority classifier discrimination. Then, a hyperspectral remote sensing image classification method based on texture features and semi-supervised learning is implemented. The LBP is employed to extract features of spatial texture information from remote sensing images and enrich the feature information of samples. The multivariate logistic regression model is used to select the unlabeled samples with the largest amount of information, and the unlabeled samples with neighborhood information and priority classifier tags are selected to obtain the pseudo-labeled samples after learning. By making full use of the advantages of sparse representation and mixed logistic regression model, a new classification model based on semi-supervised learning is constructed to effectively achieve accurate classification of hyperspectral images. The data of Indian Pines, Salinas scene and Pavia University are selected to verify the validity of the proposed method. The experiment results show that the proposed classification method obtains higher classification accuracy and shows stronger timeliness and generalization ability.*

**COMMENT 2:** In the introduction, the authors should clearly indicate the contributions and innovations of this paper.

**RESPONSE:** Thank you very much for the insightful comments. According to expert advice, we have substantially modified our manuscript in order to clearly indicate the contributions and innovations of this paper in the introduction. Please read our revised manuscript, thanks!

*The main contributions of this paper are described as follows.*

*1) A novel a hyperspectral remote sensing image classification method based on texture features and semi-supervised learning is proposed, which introduces local binary pattern, sparse representation, hybrid logistic regression model and so on.*

*2) The local binary pattern is used to effectively extract the features of spatial texture information of remote sensing images and enrich the feature information of samples.*

*3) A multiple logistic regression model was used to optimally select unlabeled samples, which are labeled by using neighbourhood information and priority classifier discrimination to achieve pseudo-labeling of unlabeled samples.*

*4) A hyperspectral remote sensing image classification model based on semi-supervised learning is constructed to effectively achieve accurate classification of hyperspectral images by making full use of the advantages of sparse representation and mixed logistic regression model.*

**COMMENT 3:** All acronyms and variables in equations must be defined in the article.

**RESPONSE:** Thank you very much for the insightful comments. According to expert advice, we have substantially modified our manuscript in order to define all acronyms and variables in equations. Please read our revised manuscript, thanks!

**COMMENT 4:** In Section 3.2, how to realize the sample labeling by using neighborhood information and priority classifier?

**RESPONSE:** Thank you very much for the insightful comments. In this study, the ground objects are closer, the correlation is stronger. In the research of sample labeling, spatial neighborhood information based on training samples is widely used. The label of any pixel on a hyperspectral image must be consistent with the label of one pixel in its neighborhood. This property can be applied to label the unlabeled samples. The label information of training samples around the unlabeled samples can be used to discriminate the unlabeled samples. The labeling discrimination method based on neighborhood information centers on the sample to be labeled. The labeled samples appearing around it are labeled with a block diagram. Then, the labeled samples are used as training samples to train the classifier and classify the unlabeled samples. Therefore, a sample labeling method based on priority classifier discrimination is proposed. For unlabeled samples with the neighborhood information, the classifier with the highest priority is used for prediction. Please read our revised manuscript, thanks!

*3.2 A sample labeling method based on neighborhood information and priority classifier*

*The features of hyperspectral images have some correlation. The ground objects are closer, the correlation is stronger. In the research of sample labeling, spatial neighborhood information based on training samples is widely used. However, due to the unknown central pixel and the lack of sufficient determination information, the neighborhood information of unlabeled samples is relatively less in the research of sample labeling. Generally, the label of any pixel on a hyperspectral image must be consistent with the label of one pixel in its neighborhood. This property can be applied to label the unlabeled samples. The label information of training samples around the unlabeled samples can be used to discriminate the unlabeled samples. The labeling discrimination method based on neighborhood information centers on the sample to be labeled. The labeled samples appearing around it are labeled with a block diagram. All the occurrences of sample labels are recorded and denoted as the neighborhood information set. Then, the labeled samples are used as training samples to train the classifier and classify the unlabeled samples. Determine whether the predicted sample label by the classifier appears in the neighborhood information set of the unlabeled samples. If it appears, the predicted label by the classifier is the sample label. Otherwise, the samples are put to be labeled back into the unlabeled sample set. One of the most important problems is whether the unlabeled samples which satisfy the neighborhood information can be reliably labeled by the classifier. At present, some studies use multiple classifiers to discriminate together and achieve good classification effect. However, a problem is how to determine the determination of labels, when the predicted labels by multiple classifiers are inconsistent, but all appear in the neighborhood information set of unlabeled samples.*

*Therefore, a sample labeling method based on priority classifier discrimination is proposed in this paper. For unlabeled samples with the neighborhood information, the classifier with the highest priority is used for prediction. If the obtained prediction marker appears in the neighborhood information set, its marker is determined. Otherwise, the classifier with the lowest priority is used for prediction. Then judge whether the label can be determined until the end of the sample labeling. The sample labeling method based on neighborhood information and priority classifier discrimination is shown in Figure 2.*

[Figure]

*Figure 2. Sample labeling process based on neighborhood information and priority classifier discrimination*
*This labeling method is a cyclical iterative process. Although it is not possible to ensure enough training samples around all unlabeled samples at the initial stage of sample labeling, it can ensure that some unlabeled samples are sufficient. The unlabeled samples are then labeled and extended to the training set. With each iteration, the training set grows. Those unlabeled samples whose neighborhood training samples are not sufficient may reach the label condition at a certain labeling time. This sample labeling method with replacement ensures the accuracy of sample labeling to a certain extent, and improves the performance of classifier step by step.*

**COMMENT 5:** Figure 2 and Figure3 are not clear, please provide some clear figures.

**RESPONSE:** Thank you very much for the insightful comments. According to expert advice, we have substantially modified our manuscript in order to provide some clear figures. Please read our revised manuscript, thanks!

| 7 | | ... | | 28 |
|---|---|---|---|---|
| | 79 | 26 | 78 | |
| | 132 | 68 | 10 | |
| | 30 | 202 | 252 | |
| 24 | | ... | | 59 |

**Figure1.** The quantized texture feature form of one region

[Figure]

**Figure 2.** Sample labeling process based on neighborhood information and priority classifier discrimination

[Figure]

**Figure 3.** Hyperspectral image classification model based on texture features and semi-supervised learning

.

[Figure]

(a)Indian Pines  (b)Pavia University  (c)Salinas Scene

**Figure 7** The classification results of the initial samples

(a)Indian Pines  (b)Pavia University  (c)Salinas Scene

**Figure 8** The classification results of the labeling samples

**COMMENT 6:** Why did you use the selected evaluation criteria? What are their advantages?

**RESPONSE:** Thank you very much for the insightful comments. Confusion Matrix (CM) is usually used in the classification and evaluation of hyperspectral images. Based on the confusion matrix, three classification indexes can be obtained, which are Overall Accuracy (OA), Average Accuracy (AA) and Kappa coefficient, which comprehensively considers the number of objects correctly classified and the error of being misclassified on the diagonal of the confusion matrix. Please read our revised manuscript, thanks!

*5.1. Evaluation criteria*

*Confusion Matrix (CM) is usually used in the classification and evaluation of hyperspectral images. A confusion matrix is generally defined as follow.*

$$P = \begin{bmatrix} p_{11} & p_{12} & \cdots & p_{1n} \\ p_{21} & p_{22} & \cdots & p_{2n} \\ \cdots & \cdots & \cdots & \cdots \\ p_{n1} & p_{n2} & \cdots & p_{nn} \end{bmatrix} \tag{14}$$

*where, n denotes the number of objects in the category, $p_{ij}$ represents the number of samples belonging to class i that were assigned to class j. The total amount of data in each row denotes the true number of objects in that category. The total amount of data for each column represents the total number of samples.*

*Based on the confusion matrix, three classification indexes can be obtained, which are Overall Accuracy (OA), Average Accuracy (AA) and Kappa coefficient.*

$$OA = \frac{\sum_{i=1}^{n} p_{ii}}{N} \tag{15}$$

*where, N represents the total number of samples participating in the classification. $p_{ii}$ represents the number of correctly classified samples of class i. It represents the probability that the classified result corresponds to its true label for each random sample.*

$$CA_i = \frac{p_{ii}}{N_i} \tag{16}$$

*where, $N_i$ represents the total number of samples for the first category in class i. $CA_i$ represents the probability that category i is correctly classified.*

$$Kappa = \frac{\left(n\left(\sum_{i=1}^{N} p_{ii}\right) - \sum_{i=1}^{N}\left(\sum_{j=1}^{N} p_{ij} \sum_{j=1}^{N} p_{ji}\right)\right)}{n^2 - \sum_{i=1}^{N}\left(\sum_{j=1}^{N} p_{ij} \sum_{j=1}^{N} p_{ji}\right)} \tag{17}$$

*The Kappa coefficient comprehensively considers the number of objects correctly classified and the error of being misclassified on the diagonal of the confusion matrix.*

**COMMENT 7:** There are some grammatical mistakes and typo errors. Please proof read from native speaker.

**RESPONSE:** Thank you very much for the insightful comments. According to expert advice, we have substantially modified our manuscript in order to eliminate a number of grammatical errors and spelling errors. In addition, we have invited an English teacher whose native language is English to check the manuscript carefully in order to improve the written English level and avoid solecism and spelling mistakes. Let the revised manuscript be more readable. Please read our revised manuscript, thanks!

**COMMENT 8:** Please add what the next work of this article is.

**RESPONSE:** Thank you very much for the insightful comments. According to expert advice, we have

substantially modified our manuscript in order to add what the next work of this article is. Please read our revised manuscript, thanks!

*However, the proposed classification method has the more computing time, so the next step should be more in-depth research to reduce the time complexity.*

**COMMENT 9:** Some new references should be added to improve the reviews the literatures.
**RESPONSE:** Thank you very much for the insightful comments. According to expert advice, we have substantially modified our manuscript in order to add some new references to improve the reviews the literatures. Please read our revised manuscript, thanks!

**And so on, please read our revised manuscript. We thank the comments and the opportunity for us to improve our manuscript. As much as possible, the questions were taken into account during the preparation of the revised manuscript. We hope that the manuscript is now suitable for publication.**

---

## Author Comment (AC3)

**Dear reviewer RC3:**

    Thank you very much for the insightful comments. Thank you for giving us a choice to correct the shortcoming of our manuscript. We already carefully read your comments and revised the manuscript according to your suggestions. We hope that this revision will make our manuscript meet the publisher. The responses to the comments point by point are listed below. Please feel free to contact us with any questions. If the revised manuscript maybe exists the shortcomings, please tell us. We will try our best to continue to revise our manuscript in order to improve our manuscript. Really thank your insightful comments and help again!

Yours sincerely,

    Regards,

    Xiangbing Zhou

**Reviewer #1:**

In this paper, local binary pattern (LBP), sparse representation and mixed logistic regression model are introduced, and a sample labeling method based on neighborhood information and priority classifier discrimination is presented. The research work reported is interesting in the community. Some suggestions are listed below to improve the manuscript's quality:

1. The manuscript's motivations should be further highlighted in the manuscript, e.g., what problems did the previous works exist? How to solve these problems? Please explain that.

2. The authors must clearly explain the difference(s) between the proposed method and similar works in the introduction.

3. The authors should further highlight the manuscript's innovations and contributions.

4. Could you tell me the limitations of the proposed method?  Please add this part to the manuscript.

5. There are a few typos and grammar errors in the manuscript. Please polish the manuscript carefully.

**COMMENT 1:** The manuscript's motivations should be further highlighted in the manuscript, e.g., what problems did the previous works exist? How to solve these problems? Please explain that.

**RESPONSE:** Thank you very much for the insightful comments. According to expert advice, we have substantially modified our manuscript in order to further highlight the manuscript's motivations in the manuscript. In this study, the hyperspectral images contain rich spectral and spatial information of earth surface features, which increases the difficulty of data processing and analysis. In addition, the training samples of actual hyperspectral images are small and there is a problem of sample labeling. It will be very difficult to solve these by using the previous methods. In this paper, local binary pattern (LBP), sparse representation and mixed logistic regression model are introduced to propose a sample labeling method based on neighborhood information and priority classifier discrimination. Then, a hyperspectral remote sensing image classification method based on texture features and semi-supervised learning is implemented to solve these problems. The LBP is employed to extract features of spatial texture information from remote sensing images and enrich the feature information of samples. The multivariate logistic regression model is used to select the unlabeled samples with the largest amount of information,

and the unlabeled samples with neighborhood information and priority classifier tags are selected to obtain the pseudo-labeled samples after learning. Please read our revised manuscript, thanks!

**COMMENT 2:** The authors must clearly explain the difference(s) between the proposed method and similar works in the introduction.

**RESPONSE:** Thank you very much for the insightful comments. According to expert advice, we have substantially modified our manuscript in order to clearly explain the difference(s) between the proposed method and similar works in the introduction. Please read our revised manuscript, thanks!

*To sum up, hyperspectral images contain rich spectral and spatial information of earth surface features, which increases the difficulty of data processing and analysis. In addition, the training samples of actual hyperspectral images are small and there is a problem of sample labeling. The local binary pattern, sparse representation and mixed logistic regression model are used in this paper. A new hyperspectral image feature extraction method based on local binary pattern is proposed to obtain texture features of hyperspectral image samples and enrich hyperspectral image sample information. A sample selection strategy based on active learning is designed to determine the unlabeled samples. Based on this, a new sample labeling method based on neighbourhood information and priority classifier discrimination is deeply studied to expand the training samples. The hyperspectral remote sensing image classification method based on texture features and semi-supervised learning is studied to improve the classification accuracy of remote sensing images.*

*The main contributions of this paper are described as follows.*

*1) A novel a hyperspectral remote sensing image classification method based on texture features and semi-supervised learning is proposed, which introduces local binary pattern, sparse representation, hybrid logistic regression model and so on.*

*2) The local binary pattern is used to effectively extract the features of spatial texture information of remote sensing images and enrich the feature information of samples.*

*3) A multiple logistic regression model was used to optimally select unlabeled samples, which are labeled by using neighbourhood information and priority classifier discrimination to achieve pseudo-labeling of unlabeled samples.*

*4) A hyperspectral remote sensing image classification model based on semi-supervised learning is constructed to effectively achieve accurate classification of hyperspectral images by making full use of the advantages of sparse representation and mixed logistic regression model.*

**COMMENT 3:** The authors should further highlight the manuscript's innovations and contributions.

**RESPONSE:** Thank you very much for the insightful comments. According to expert advice, we have substantially modified our manuscript in order to further highlight the manuscript's innovations and contributions.. Please read our revised manuscript, thanks!

*The main contributions of this paper are described as follows.*

*1) A novel a hyperspectral remote sensing image classification method based on texture features and semi-supervised learning is proposed, which introduces local binary pattern, sparse representation, hybrid logistic regression model and so on.*

*2) The local binary pattern is used to effectively extract the features of spatial texture information of remote sensing images and enrich the feature information of samples.*

*3) A multiple logistic regression model was used to optimally select unlabeled samples, which are labeled by using neighbourhood information and priority classifier discrimination to achieve pseudo-labeling of unlabeled samples.*

*4) A hyperspectral remote sensing image classification model based on semi-supervised learning is constructed to effectively achieve accurate classification of hyperspectral images by making full use of the advantages of sparse representation and mixed logistic regression model.*

**COMMENT 4:** Could you tell me the limitations of the proposed method? Please add this part to the manuscript.

**RESPONSE:** Thank you very much for the insightful comments. According to expert advice, we have substantially modified our manuscript in order to the limitations of the proposed method. In this study, the proposed classification method has the more computing time, so the next step should be more in-depth research to reduce the time complexity. Please read our revised manuscript, thanks!

**COMMENT 5:** There are a few typos and grammar errors in the manuscript. Please polish the manuscript carefully.

**RESPONSE:** Thank you very much for the insightful comments. According to expert advice, we have substantially modified our manuscript in order to eliminate a number of grammatical errors and spelling errors. In addition, we have invited an English teacher whose native language is English to check the manuscript carefully in order to improve the written English level and avoid solecism and spelling mistakes. Let the revised manuscript be more readable. Please read our revised manuscript, thanks!

**And so on, please read our revised manuscript. We thank the comments and the opportunity for us to improve our manuscript. As much as possible, the questions were taken into account during the preparation of the revised manuscript. We hope that the manuscript is now suitable for publication.**

---

## Author Comment (AC4)

**Dear reviewer RC4:**

Thank you very much for the insightful comments. Thank you for giving us a choice to correct the shortcoming of our manuscript. We already carefully read your comments and revised the manuscript according to your suggestions. We hope that this revision will make our manuscript meet the publisher. The responses to the comments point by point are listed below. Please feel free to contact us with any questions. If the revised manuscript maybe exists the shortcomings, please tell us. We will try our best to continue to revise our manuscript in order to improve our manuscript. Really thank your insightful comments and help again!

Yours sincerely,

Regards,

Xiangbing Zhou

**Reviewer #RC4:**

The proposal is interesting, and the experiments conducted are good, however the manuscript presents many flaws in its present form.

1.In the abstract, it is better to improve some sentences and the innovation and achievement of the paper that comprise it from other similar work is ambiguous and is better to add to the abstract. The introduction is not organized well and the lack of consistency in the story's narration is apparent. Please improve it if possible. Some modifications in terms of eliminating such general and clear information are needed.

2. Literature review is a little bit insufficient, please add more recent good works by researchers..

3. The quality of the figure should be improved as much as possible. For instance, in Fig.1,...

4. What are the limitations behind this study? This topic should be highlighted in the Conclusion of manuscript.

5. The conclusions should be more concrete with data. Please improve them.

**COMMENT 1:** T In the abstract, it is better to improve some sentences and the innovation and achievement of the paper that comprise it from other similar work is ambiguous and is better to add to the abstract. The introduction is not organized well and the lack of consistency in the story's narration is apparent. Please improve it if possible. Some modifications in terms of eliminating such general and clear information are needed.

**RESPONSE:** Thank you very much for the insightful comments. According to expert advice, we have substantially modified our manuscript in order to improve some sentences and the innovation and achievement of the paper that comprise it from other similar work is ambiguous and is better to add to the abstract. In addition, The introduction is reorganized and the consistency in the story's narration is improved. Some modifications in terms of eliminating such general and clear information are added in our revised paper. Please read our revised manuscript, thanks!

*Abstract:Hyperspectral images contain abundant spectral and spatial information of the surface of earth, which increase the difficulties of data processing and analysis, and sample labeling. In this paper, local binary pattern (LBP),*

*sparse representation and mixed logistic regression model are introduced to propose a sample labeling method based on neighborhood information and priority classifier discrimination. Then, a hyperspectral remote sensing image classification method based on texture features and semi-supervised learning is implemented. The LBP is employed to extract features of spatial texture information from remote sensing images and enrich the feature information of samples. The multivariate logistic regression model is used to select the unlabeled samples with the largest amount of information, and the unlabeled samples with neighborhood information and priority classifier tags are selected to obtain the pseudo-labeled samples after learning. By making full use of the advantages of sparse representation and mixed logistic regression model, a new classification model based on semi-supervised learning is constructed to effectively achieve accurate classification of hyperspectral images. The data of Indian Pines, Salinas scene and Pavia University are selected to verify the validity of the proposed method. The experiment results show that the proposed classification method obtains higher classification accuracy and shows stronger timeliness and generalization ability.*

**COMMENT 2:** Literature review is a little bit insufficient, please add more recent good works by researchers..

**RESPONSE:** Thank you very much for the insightful comments. According to expert advice, we have substantially modified our manuscript in order to add more recent good works by researchers, which can improve the reviews the literatures. Please read our revised manuscript, thanks!

*Camps-Valls G, Bandos T, Zhou D. (2007). Semi-supervised graph-based hyperspectral image classification. IEEE Transactions on Geoscience and Remote Sensing, 45(10):3044-3054.*

*Chang C, Kuo Y, Chen S, Liang C, Ma KY, Hu PF. (2021). Self-Mutual information-based band selection for hyperspectral image classification. IEEE Trans Geoscience and Remote Sensing, 59 (7): 5979-5997.*

*Chen C, Ma Y, Ren G. (2020) .Hyperspectral classification using deep belief networks based on conjugate gradient update and pixel-centric spectral block features. IEEE Journal of Selected Topics in Applied Earth Observations and Remote Sensing, 13:4060-4069.*

*Chen GY. (2021). Multiscale filter-based hyperspectral image classification with PCA and SVM. Journal of Electrical Engineering, 72(1):pp. 40-45*

*Chen, HY, Fang M, Xu S. (2020). Hyperspectral remote sensing image classification with CNN based on quantum genetic-optimized sparse representation. IEEE Access, 8: 99900-99909.*

*Chen H, Miao F, Chen Y. (2021). A hyperspectral image classification method using multifeature vectors and optimized KELM. IEEE Journal of Selected Topics in Applied Earth Observations and Remote Sensing, 14: 2781-2795.*

*Chen Y, Nasser MN, Tran TD. (2011). Hyperspectral image classification using dictionary-based sparse representation. IEEE Transactions on Geoscience and Remote Sensing, 49(10):3973-3985.*

*Chen Y, Nasser MN, Tran TD. (2013). Hyperspectral image classification via kernel sparse representation. IEEE Transactions on Geoscience and Remote Sensing, 51(1):217–231.*

*Cui M, Prasad S. (2013).Multiscale sparse representation classification for robust hyperspectral image analysis. IEEE Global Conference on Signal and Information Processing, 969-972.*

*Deng W, Zhang L, Zhou X, et al. (2022). Multi-strategy particle swarm and ant colony hybrid optimization for airport taxiway planning problem. Information Sciences, 612: 576-593.*

*Duan Z, Song P, Yang C, et al. (2022). The impact of hyperglycaemic crisis episodes on long-term outcomes for inpatients presenting with acute organ injury: A prospective, multicentre follow-up study. Frontiers in Endocrinology, Doi: 10.3389/fendo.2022.1057089*

*Dou Z, Gao K, Zhang X, Wang H, Han L. (2020). Band selection of hyperspectral images using attention-based autoencoders. IEEE Geoscience and Remote Sensing Letters, 18 (1): 147-151.*

Dumke I, Ludvigsen M, Ellefmo SL, Søreide F, Johnsen G, Murton B. (2019). Underwater hyperspectral imaging using a stationary platform in the transatlantic geotraverse hydrothermal field. IEEE Transactions on Geoscience and Remote Sensing, 57 (5): 2947-2962.

Huang C, Zhou X, Ran X, et al.(2023). Co-evolutionary competitive swarm optimizer with three-phase for large-scale complex optimization problem. Information Sciences, 619:2-18.

Huang C, Zhou X, Ran X, et al.(2023). Adaptive cylinder vector particle swarm optimization with differential evolution for UAV path planning . Engineering Applications of Artificial Intelligence,121:105942.

Huang W, Huang Y, Wang H, Liu Y, Shim HJ. (2020). Local binary patterns and superpixel-based multiple kernels for hyperspectral image classification. IEEE Journal of Selected Topics in Applied Earth Observations and Remote Sensing, 13: 4550-4563.

Hu S, Xu C, Peng J, Yan X, Long T. (2019).Weighted Kernel joint sparse representation for hyperspectral image classification. IET Image Processing, 13(2):254-260.

Jiang X, Liu W, Zhang Y, Liu J, Li S, Lin J. (2020). Spectral-spatial hyperspectral image classification using dual-channel capsule networks. IEEE Geoscience and Remote Sensing Letters, 18 (6): 1094-1098.

Liu ZX, Ma L, Du Q. (2021).Class-wise distribution adaptation for unsupervised classification of hyperspectral remote sensing images. IEEE Transactions on Geoscience and Remote Sensing, 59(1): 508-521

Melgani F, Bruzzone L. (2004). Classification of hyperspectral remote sensing images with support vector machines. 2004. IEEE Transactions on Geoscience and Remote Sensing, 42(8):1778-1790.

Ojala T, Harwood I. (1996).A comparative study of texture measures with classification based on feature distributions. Pattern Recognition29(1):51-59.

Ratle, Terrettaz-Zufferey, Kanevski, et al. (206). Learning manifolds in forensic data. international conference on artificial neural networks. Springer, Berlin, Heidelberg.

Samiappan S, Moorhead R J. (2015). Semi-supervised co-training and active learning framework for hyperspectral image classification.2015 IEEE International Geoscience and Remote Sensing Symposium(IGARSS), IEEE:401-404.

Seifi M, Ghassemian H. (2017). A probabilistic SVM approach for hyperspectral image classification using spectral and texture features. International Journal of Remote Sensing, 38 (15): 4265-4284.

Shang X, Song M, Chang CI. (2020). An iterative random training sample selection approach to constrained energy minimization for hyperspectral image classification. IEEE Geoscience and Remote Sensing Letters, 18 (9): 1625-1629.

Shi C, Pun CM. (2019). Multiscale superpixel-based hyperspectral image classification using recurrent neural networks with stacked autoencoders. IEEE Transactions on Multimedia, 22 (2): 487-501.

Song Y, Cai X, Zhou X, et al. (2022). Dynamic hybrid mechanism-based differential evolution algorithm and its application. Expert Systems with Applications, 213: 118834.

Song Y, Zhao G, Zhang B, et al.(2023). An enhanced distributed differential evolution algorithm for portfolio optimization problems. Engineering Applications of Artificial Intelligence,121:106004

Tan K, Li E, Qian D, et al. (2014). An efficient semi-supervised classification approach for hyperspectral imagery. ISPRS Journal of Photogrammetry & Remote Sensing, 97:36-45.

Tang YY, Yuan H, Li L. (2014).Manifold-based sparse representation for hyperspectral image classification. IEEE Transactions on Geoscience and Remote Sensing, 52(12):7606-7618.

Wang C,Wang H,Hu B,Jia W,Xu J,Li X. (2016). A novel spatial-spectral sparse representation for hyperspectral image classification based on neighborhood segmentation. Spectroscopy and Spectral Analysis, 36(9):2919-2924.

Wang HR, Celik T. (2018). Sparse representation-based hyperspectral image classification. Signal Image and Video Processing, 12(5):1009-1017.

Wang QY, Zhang Q, Zhang JP, Kang SQ, Wang YJ. (2022). Graph-based semisupervised learning with weighted features for hyperspectral remote sensing image classification. IEEE Journal of Selected Topics in Applied Earth Observations and Remote Sensing, 15: 6356-6370

Xu J.; Zhao Y.; Chen H.; Deng W.(2023). ABC-GSPBFT: PBFT with grouping score mechanism and optimized consensus process for flight operation data-sharing. Information Sciences, 624:110-127.

Xue ZH, Du PJ, Li J, Su HJ. (2017). Sparse graph regularization for hyperspectral remote sensing image classification. IEEE Transactions on Geoscience and Remote Sensing, 55(4): 2351-2366

*Yang C, Liu S C, Bruzzone L, et al. (2012). A semisupervised feature metric-based band selection method for hyperspectral image classification. Hyperspectral Image and Signal Processing (WHISPERS), 2012 4th Workshop on. IEEE.*

*Yang M, Li CH, Guan J, Yan XS. (2018). A supervised-learning p-norm distance metric for hyperspectral remote sensing image classification. IEEE Geoscience and Remote Sensing Letters, 15(9): 1432-1436*

*Yang X, Cao W, Lu Y, et al. (2022). Hyperspectral image transformer classification networks. IEEE Transactions on Geoscience and Remote Sensing 60: 1- 15. doi:10.1109/TGRS.2022.3171551*

*Ye X, Ma J, Xiong H. (2021). Local affine preservation with motion consistency for feature matching of remote sensing images. IEEE Transactions on Geoscience and Remote Sensing, 60: 1-12.*

*Yin J, Qi C, Chen Q, Qu J. (2021). Spatial-spectral network for hyperspectral image classification: A 3-D CNN and Bi-LSTM framework. Remote Sensing, 13 (12), 2353.*

*Yu C, Liu C, Yu H, Song M, Chang CI.(2021). Unsupervised domain adaptation with dense-based compaction for hyperspectral imagery..IEEE Journal of Selected Topics in Applied Earth Observations and Remote Sensing 14: 12287–12299.*

*Yu C, Zhou S, Song M, Chang CI. (2021). Semisupervised hyperspectral band selection based on dual-constrained low-rank representation. IEEE Geoscience and Remote Sensing Letters, 19: 1-5.*

*Zhang CJ, Li GD, Du SH. (2019). Multi-scale dense networks for hyperspectral remote sensing image classification. IEEE Transactions on Geoscience and Remote Sensing, 57(11): 9201-9222*

*Zhang J, Meng Z, Zhao F, Liu H, (2022). Chang Z. Convolution transformer mixer for hyperspectral image classification. IEEE Geoscience and Remote Sensing Letters, doi:10.1109/LGRS.2022.3208935.*

*Zhao H, Wang C, Chen H, Chen T, Deng W.(2023). A hybrid classification method with dual-channel CNN and KELM for hyperspectral*

*remote sensing images, International Journal of Remote Sensing, 44(1):289-310*

*Zhang X, Wang H, Du C, et al.(2022). Custom-molded offloading footwear effectively prevents recurrence and amputation, and lowers mortality rates in high-risk diabetic foot patients: a multicenter, prospective observational study. Diabetes, Metabolic Syndrome and Obesity: Targets and Therapy, 15: 103-109*

*Zhao XD, Zhang MM, Tao R, Li W, Liao WZ, Tian LF, Philips W. (2022). Fractional Fourier image transformer for multimodal remote sensing data classification. IEEE Transactions on Neural Networks and Learning Systems, doi: 10.1109/TNNLS.2022.3189994*

*Zhang Z, Crawford M. (2016). Semi-supervised multi-metric active learning for classification of hyperspectral images.2016 IEEE International Geoscience and Remote Sensing Symposium(IGARSS), IEEE:1843-1847.*

*Zhao X, Zhang M, Tao R, et al.(2022). Fractional Fourier image transformer for multimodal remote sensing data classification. IEEE Transactions on Neural Networks and Learning Systems 1–13. doi:10.1109/TNNLS.2022.3189994.*

*Zhong K, Zhou G, Deng W, et al. (2021). MOMPA: Multi-objective marine predator algorithm. Computer Methods in Applied Mechanics and Engineering, 385:114029.*

*Zhou S, Xue Z, Du P. (2019). Semisupervised stacked autoencoder with cotraining for hyperspectral image classification.*
*IEEE Transactions on Geoscience and Remote Sensing 57 (6): 3813–3826.*

**COMMENT 3:** The quality of the figure should be improved as much as possible. For instance, in Fig.1,..

**RESPONSE:** Thank you very much for the insightful comments. According to expert advice, we have substantially modified our manuscript in order to improve the quality of the figures. Please read our revised manuscript, thanks!

| 7 | | ... | | 28 |
|---|---|---|---|---|
| | 79 | 26 | 78 | |
| | 132 | 68 | 10 | |
| | 30 | 202 | 252 | |
| 24 | | ... | | 59 |

**Figure1.** The quantized texture feature form of one region

[Figure]

**Figure 2.** Sample labeling process based on neighborhood information and priority classifier discrimination

[Figure]

**Figure 3.** Hyperspectral image classification model based on texture features and semi-supervised learning

.

[Figure]

[Figure]

[Figure]

(a)Indian Pines         (b)Pavia University        (c)Salinas Scene

**Figure 7** The classification results of the initial samples

[Figure]

(a)Indian Pines         (b)Pavia University        (c)Salinas Scene

**Figure 8** The classification results of the labeling samples

**COMMENT 4:** What are the limitations behind this study? This topic should be highlighted in the Conclusion of manuscript.

**RESPONSE:** Thank you very much for the insightful comments. In this stud, the proposed classification method has the more computing time, so the next step should be more in-depth research to reduce the time complexity. Therefore, this topic has highlighted in the Conclusion of manuscript. Please read our revised manuscript, thanks!

*5. Conclusion*

*However, the proposed classification method has the more computing time, so the next step should be more in-depth research to reduce the time complexity.*

**COMMENT 5:** The conclusions should be more concrete with data. Please improve them.

**RESPONSE:** Thank you very much for the insightful comments. According to expert advice, we have substantially modified our manuscript in order to add more concrete with data in the conclusions. Please read our revised manuscript, thanks!

*6. Conclusion*

*For the difficulties of hyperspectral image processing and analysis, a new sample labeling method based on neighborhood information and priority classifier discrimination is developed to implement a new hyperspectral remote sensing image classification method based on texture features and semi-supervised learning by introducing local binary model, sparse representation and mixed logistic regression model. The local binary pattern is employed to deal with the hyperspectral data and extract the texture features of the hyperspectral remote sensing image. The multivariate logistic regression model is used to select the unlabeled samples with the largest amount of information, and the unlabeled samples with neighborhood information and priority classifier tags are selected to obtain the pseudo-labeled samples after learning. The problem of limited labeled samples of hyperspectral images is solved. The data of Indian*

*Pines, Salinas scene and Pavia University are selected in here. The experiment results of the BT method are obviously better than those of other methods. The block window of Indian Pines dataset is 7\*7. The block windows of Pavia University and Salinas scene are 25 \* 25 and 20 \* 20, respectively. The combination of MLR and SRC can get better classification results. The obtained classification results by the classifier and the labeled samples are smoother and has fewer discrete points, which indicates that the generalization ability of the classifier is improved by labeling the samples from the classification visualization. For Indian Pines data, the classification results of AA, OA and KAPPA are 84.7%, 94.42% and 0.914, respectively. For Pavia University data, the classification results of AA, OA and KAPPA are 81.87%, 88.53% and 0.848, respectively. For Salinas Scene data, the classification results of AA, OA and KAPPA are 87.76%, 92.64% and 0.918, respectively. Therefore, the classification method obtains the higher classification accuracy.*

*However, the proposed classification method has the more computing time, so the next step should be more in-depth research to reduce the time complexity.*

**And so on, please read our revised manuscript. We thank the comments and the opportunity for us to improve our manuscript. As much as possible, the questions were taken into account during the preparation of the revised manuscript. We hope that the manuscript is now suitable for publication.**